

**Seasonal provenance changes of present-day Saharan dust**
**collected on- and offshore Mauritania**
Carmen A. Friese[1], Hans van Hateren[2,*], Christoph Vogt[1,3], Gerhard Fischer[1], Jan-Berend W.
Stuut[1,2]
[1]Marum-Center of Marine Environmental Sciences, University of Bremen, Bremen, 28359, Germany
[2]NIOZ-Royal Netherlands Institute for Sea Research, Department of Ocean Systems, and Utrecht University, 1790
AB, Den Burg Texel, Netherlands
[3]ZEKAM, Crystallography, Geosciences, University of Bremen, 28359, Germany
[*]Now at: Vrije Universiteit Amsterdam, Faculty of Earth Sciences, 1081 HV Amsterdam, the Netherlands
*Correspondence to*: Carmen A. Friese (cfriese@marum.de)
**Abstract.**
Saharan dust has a crucial influence on the earth climate system and its emission, transport, and deposition are
intimately related to environmental parameters. The alteration in the physical and chemical properties of Saharan
dust due to changes in environmental parameters is often used to reconstruct the climate of the past. However, to
better interpret possible climate changes the dust source regions need to be known. By analysing the mineralogical
composition of transported or deposited dust, potential dust source areas can be inferred. Summer dust transport
offshore Northwest Africa occurs in the Saharan air layer (SAL). In contrast, dust transport in continental dust
source areas occurs predominantly with the trade winds. Hence, the source regions and related mineralogical
tracers differ with season and sampling location. To test this, dust collected in traps onshore and in oceanic
sediment traps offshore Mauritania during 2013 to 2015 was analysed. Meteorological data, particle-size
distributions, back-trajectory and mineralogical analyses were compared to derive the dust provenance and
dispersal. For the onshore dust samples, the source regions varied according to the seasonal changes in trade-wind
direction. Gibbsite and dolomite indicated a Western Saharan and local source during summer, while chlorite,
serpentine and rutile indicated a source in Mauritania and Mali during winter. In contrast, for the samples that were
collected offshore, dust sources varied according to the seasonal change in the dust transporting air layer. In
summer, dust was transported in the SAL from Mauritania, Mali and Libya as indicated by ferryglaucophane and
zeolite. In winter, dust was transported with the Trades from the Western Sahara as indicated by e.g. sepiolite and
fluellite.
**Keywords**
Saharan dust, MWAC, sediment trap, mineralogy, particle size, major potential source area, provenance





**1 Introduction**
Mineral dust influences global climate through many feedback mechanisms and is in turn influenced by variations
in environmental parameters. The emission, transport and deposition of mineral dust reacts sensitively to
parameters of climate change like rainfall, wind, temperature and vegetation cover (Knippertz and Stuut, 2014).
In turn, the emission, transport and deposition of mineral dust has an impact on the atmospheric energy balance
(Haywood and Boucher, 2000), precipitation distribution and amplitude (Yoshioka et al., 2007), sea surface
temperatures (Lau and Kim, 2007) as well as the oceanic carbon pump  (Martin, 1990;Martin et al., 1991;Jickells
et al., 2005;Ploug et al., 2008a;Iversen et al., 2010;Iversen and Robert, 2015). The sensitivity of mineral dust to
environmental parameters is used to reconstruct the climate of the past (Diester-Haass and Chamley, 1978;Stein,
1985;Rea, 1994;Holz et al., 2007;Tjallingii et al., 2008;Mulitza et al., 2010). For instance, the particle size of
mineral dust in ocean sediment records varies according to the paleo-frequency of dust-storm and rainfall events
(e.g., (Friese et al., 2016). Further, the mineralogical composition of mineral dust in sediment core records can be
used as a qualitative proxy for the paleo-dust source activity (Scheuvens et al., 2013).
Every year, about 2000Mt dust are emitted from source areas around the world, of which 75% are deposited on
land and 25% into the oceans (Shao et al., 2011).The Saharan Desert is the world's largest source of mineral
aerosols with an annual dust transport of ~180 Mt westwards towards the North Atlantic (Yu et al., 2015). About
140Mt is actually deposited into the North Atlantic Ocean (Yu et al., 2015). In addition, about 430 Mt is blown
from the Sahara towards the equatorial Atlantic (Shao et al., 2011) and therefore constitutes an essential component
of the global climate system. The source regions of Saharan dust have been studied frequently by analysing the
mineralogical composition of dust collected at continental sites (e.g.(Schütz and Sebert, 1987;Khiri et al.,
2004;Kandler et al., 2009;Skonieczny et al., 2011;Skonieczny et al., 2013), during aircraft flights (e.g.(Formenti
et al., 2008), on research ships (Chester et al., 1971;Chester and Johnson, 1971a;Chester and Johnson,
1971b;Chester et al., 1972;Aston et al., 1973;Stuut et al., 2005) and with gravity cores offshore NW Africa
(Biscaye, 1964;Biscaye, 1965;Griffin et al., 1968;Rateev et al., 1969;Diester-Haass and Chamley, 1978;Lange,
1982;Meyer et al., 2013). Continental dust studies in northern Morocco revealed that dust is produced
predominantly locally (Khiri et al., 2004;Kandler et al., 2009). For instance, a high percentage of quartz and
feldspar and a low amount of micas in the dust samples was interpreted to represent mostly local dust sources and
the availability of calcite sources from proximal coastal dunes in Morocco (Khiri et al., 2004). Further, also in
Morocco, dust was sampled in Tinfou at a height of 4 m during the SAMUM 2006 field campaign. These samples
were analysed for their physical and chemical properties. The particle size correlated to local surface wind speed
suggesting the contribution of local dust (Kandler et al., 2009). In contrast, in coastal Senegal dust is sourced by
the Sahel during winter as shown by low illite/kaolinite (I/K) ratios and lower palygorskite contents as opposed to
the summer samples which were suggested to be originating from the Sahara (Skonieczny et al., 2013). Further,
the I/K ratio in dust sampled on the Cape Verde Islands showed that dust was derived from strongly varying
sources: north-western  Sahara, central and southern Sahara and the Sahel (Caquineau et al., 2002). Hence, dust
collected on land is predominantly of local provenance, while the sources of dust sampled offshore NW Africa are
of regional and long-distance provenance. As a result, a large seasonal difference can be expected in the
composition of the marine climate archives, related to the different dominating transport mechanisms of dust in
summer and winter (Friese et al., 2016).





To test this, we compared the mineralogical composition, the fluxes, and the particle size of Saharan dust sampled
from 2013-2015 in Iwik (Mauritania) in on-land dust traps with Saharan dust sampled from 2013-2015 offshore
Cape Blanc (Mauritania) in sub-marine sediment traps. By comparing the data with meteorological data, back
trajectories, the African lithology and satellite images we aim to address the following questions:
1) Is there a seasonal variation in the transport patterns of dust deposited on land?
2) What are the source regions of dust trapped on land versus dust trapped in the ocean?
3) Do these source regions vary seasonally?
4) Can we identify characteristic minerals that constitute a tracer for certain source areas?
**1.1 North African dust sources**
The major potential source areas (PSA) of northern African dust are summarized in a review by Scheuvens et al.
(2013), see Fig. 1). Predominant dust transport towards western Africa and offshore the Atlantic Ocean occurs
from the foothills of the Atlas mountains, Western Sahara and Western Mauritania (**PSA 2**), southern Algeria and
northern Mali (**PSA 3**) and Western Chad including the Bodélé depression (**PSA 5**) (Scheuvens et al., 2013). In
contrast, dust sourced from Tunisia and northern Algeria (**PSA 1**) is transported predominantly to the western
Mediterranean and Western Europe (Stuut et al., 2009). Central Libya (**PSA 4**) is the most important region for
dust transport to the eastern Mediterranean (Scheuvens et al., 2013).



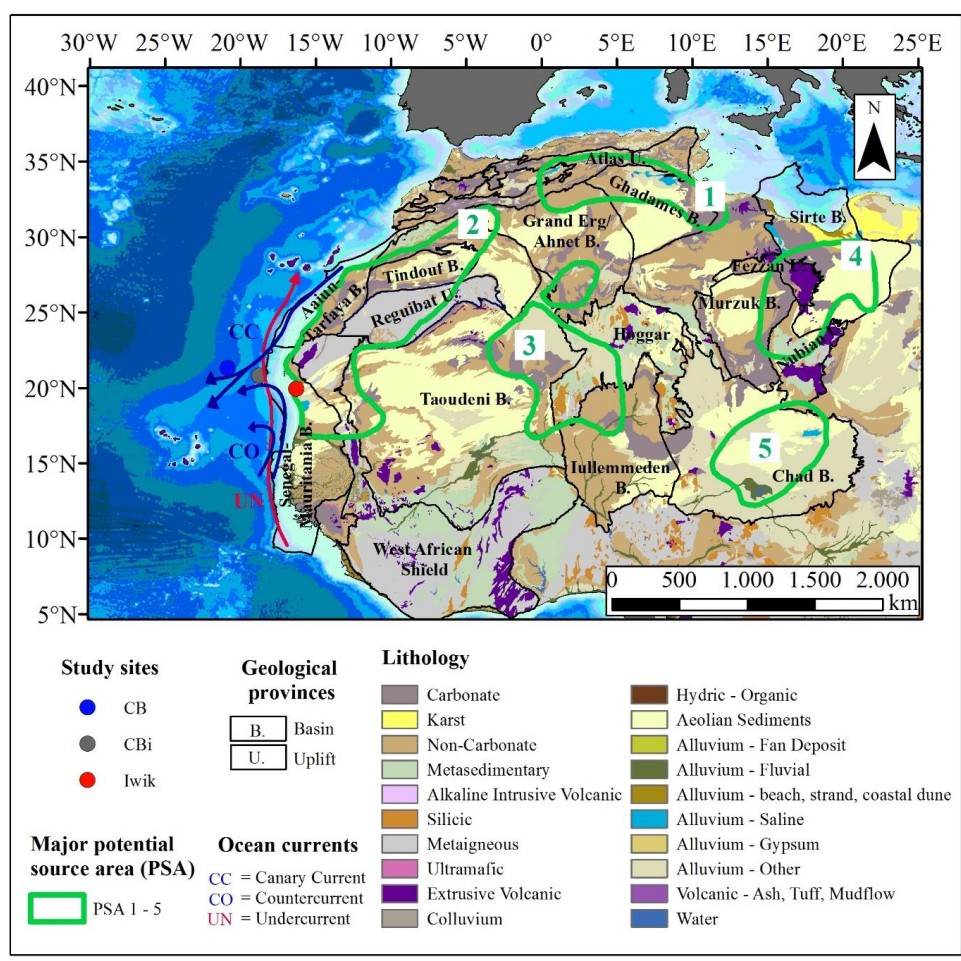


**Figure 1: Location of the sediment trap moorings CB and CBi offshore Cape Blanc and the MWAC dust collector on**

**shore near Iwik (shapefile of the surface lithology and the geological provinces: downloaded from the USGS website**

**http://rmgsc.cr.usgs.gov/ecosystems/africa.shtml#SL** **and**

**http://certmapper.cr.usgs.gov/geoportal/catalog/main/home.page, major potential dust source areas: redrawn from**

**Scheuvens et al. (2013), ocean currents: redrawn from Mittelstaedt (1991)).**

**1.2 Geological characterisation of dust-producing areas**

In the following the lithology of the geological provinces that underlay the major PSA's is outlined (Fig. 1).

The **PSA 1** is underlain by the eastern Atlas chain and the northern Grand erg/Ahnet and Ghadames Basins. The

outcrops in the Atlas uplift are composed of e.g. limestones, sandstones and evaporites (Piqué, 2001). The thick

strata overlying the northern Ahnet and Ghadames Basin consist of e.g. sandstones and mudstones (Selley, 1997a).

The **PSA 2** is underlain by the Reguibat Shield, the Mauritanides and the Senegal-Mauritania, Aaiun-Tarfaya,

Tindouf and Taoudeni Basins. The western part of the Reguibat Shield is dominated by granitic rocks, while the



eastern part is dominated by metamorphic and granitic rocks (Schofield et al. (2006) and references therein). West
of the Reguibat Shield, the Mauritanides consist of a metamorphic belt and ophiolite (Villeneuve, 2005). West of
the Taoudeni Basin, the Mauritanides are characterized by granites, quartzites and strongly metamorphosed rocks
(Villeneuve, 2005). While the Aaiun-Tarfaya Basin features outcrops with dolomites and limestones, the Senegal-
Mauritania Basin is characterized by very few carbonate deposits (Bosse and Gwosdz, 1996). The Tindouf Basin
is characterized by mainly sandy deposits (Selley, 1997b, a).
The **PSA 3** is underlain by the western Hoggar and parts of the Ahnet, Taoudeni and Iullemeden Basins. The
Pharusian belt located in the western Hoggar is characterized by Eburnean granulites, gneiss, graywackes and
magmatic rocks (Boullier, 1991). In the southern Ahnet Basin sandstone strata crop out. On the eastern edge of
the Taoudeni Basin outcropping sediments are characterized by conglomerates, sandstones and limestones
(Bertrand-Sarfati et al., 1991). The outcrops of the Iullemeden Basin are composed of e.g. sandstones,
carbonaceous shale, laterites and massive clays (Kogbe, 1973).
The **PSA 4** is underlain by parts of the Fezzan and Nubian uplifts and the Sirte and Murzuk Basins. The eastern
Fezzan uplift consists of ocean island basalts (Cvetković et al., 2010;Abdel-Karim et al., 2013), while sediments
outcropping in the northern Nubian uplift are composed of e.g. sandstones, limestones and gypsiferous horizons
(El Makkrouf, 1988). The southern Sirte Basin is covered by sands, gravel and sand seas (Selley, 1997c). Outcrops
of the eastern Murzuk Basin are composed of marine limestones and alluvial sandstones (Selley, 1997a, b).
The **PSA 5** is underlain by the Chad Basin. During the Holocene, the Chad Basin was filled with fine-grained
particles from the drainage of the Tibesti mountains to the north (Prospero et al., 2002). Hence, the sediments that
outcrop in the central Chad Basin are characterized by fluvial and alluvial sediments such as laminated diatomites,
pelites and coastal sandridges (Schuster et al., 2009).
The continental dust collector Iwik (~19°53' N, ~16° 18' W) is located in **PSA 2** in the Parc National de Banc
d'Arguin (PNBA) near Iwik in Mauritania (Fig. 1). The local soils surrounding the dust collector are composed of
sandy deposits often rich in fossil shells and partly cemented by lime (Einsele et al., 1974).
**1.3 Atmospheric setting**
Saharan dust emission, transport and deposition are related to seasonal variations in atmospheric circulation
(Knippertz and Todd, 2012). The intertropical convergence zone (ITCZ) shifts meridionally from ~12 °N during
boreal winter to ~ 21 °N during boreal summer resulting in a seasonal change in rainfall and winds over the African
continent (Nicholson, 2009).
During summer, continental rainfall is most intense and the rain belt is positioned near ~10°N with smaller amounts
of rainfall near ~ 21°N. Dust emission is driven by low level jets, so-called 'haboobs', African easterly waves
(AEWs) and high surface winds associated with the Saharan heat low (Knippertz and Todd, 2012). N trade winds
blow in coastal Mauritania year-round (National Geospatial-Intelligence Agency, 2006). The offshore transport of
Saharan dust particles occurs within the 'Saharan air layer' (SAL) at an altitude of about 3 km (Prospero and
Carlson, 1970;Carlson and Prospero, 1972;Prospero and Carlson, 1972;Diaz et al., 1976).
During winter, dust emission is driven by the break-down of nocturnal low-level jets after sunrise, increased surges
in Harmattan winds and microscale dust devils and dust plumes (Koch and Renno, 2005;Knippertz and Todd,



2012). Dust is transported within the low-level NE and E trade winds to coastal Mauritania (Dobson, 1781) and
also offshore to the sediment-trap mooring sites (Stuut et al., 2005).
**1.4 Oceanic setting**
The surface-water circulation offshore Cape Blanc is influenced by the southward-flowing Canary Current (CC)
and the poleward-flowing coastal counter current or Mauritania Current (Fig. 1). Underneath, the undercurrent is
flowing poleward in water depths down to 1000 m (Fig. 1). The undercurrent flows along the continental slope
and transports water masses originating from ~5-10 °N to latitudes up to 26 °N. The poleward flowing South
Atlantic Central Water (SACW) and the southward flowing North Atlantic Central Water (NACW) are situated
below the counter current and meet offshore Cape Blanc (Mittelstaedt, 1991). The study area is positioned in a
zone of permanent annual upwelling of sub-surface water masses (Cropper et al., 2014). The NACW and SACW
may be upwelled and mixed laterally off Cape Blanc (Meunier et al., 2012). The permanent annual upwelling of
nutrient-rich subsurface waters results in high phytoplankton concentrations offshore Cape Blanc (Van Camp et
al., 1991). As a result, the surface waters are rich in organic detritus, usually referred to as 'marine snow', and
faecal pellets which are produced by marine zooplankton (Iversen et al., 2010).

Individual Saharan dust particles which settle at the ocean surface hardly settle to the deep sea. Instead, fine dust
particles can be transferred from the ocean surface to the deep sea by being incorporated into marine snow
aggregates and faecal pellets (Ternon et al., 2010). The aggregate formation and ballasting of marine snow
aggregates and faecal pellets with marine carbonate and opal as well as with Saharan dust particles results in
anomalously high sinking velocities (Ploug et al., 2008b;Fischer and Karakas, 2009;Iversen et al., 2010;Iversen
and Ploug, 2010;Iversen and Robert, 2015). Dust-loaded particles that sink into the deeper water column are
assumed to have a mean settling speed of ~ 240 m d$^{-1}$ at site CB (Fischer and Karakas, 2009).

The buoy Carmen (~21°15' N, ~20°56' W) and the sediment trap mooring sites CB (~21°16' N, ~20°48' W) and
CBi (~20°45' N, ~18°42' W) are located ~ 200 and ~ 80 nautical miles offshore Cape Blanc in the north-eastern
(NE) equatorial Atlantic ocean (Fig. 1).


















## 2. Material and Methods

### 2.1 Sediment traps

Saharan dust was collected in the ocean using marine sediment traps of the type Kiel (model SMT-234/243) which are conical with an opening of 0.5 m² (Fig. 2). The principle of particle collection is much the same as described by Van der Does et al. (2016b) and Korte et al. (2016). At the top of the opening a honeycomb grid is installed to prevent large swimmers (>1 cm) from entering the trap. The sediment traps were equipped with twenty sample cups which rotated according to a pre-programmed sampling interval (Fischer and Wefer, 1991). The sampling interval was chosen depending on the timing of the ship expeditions.

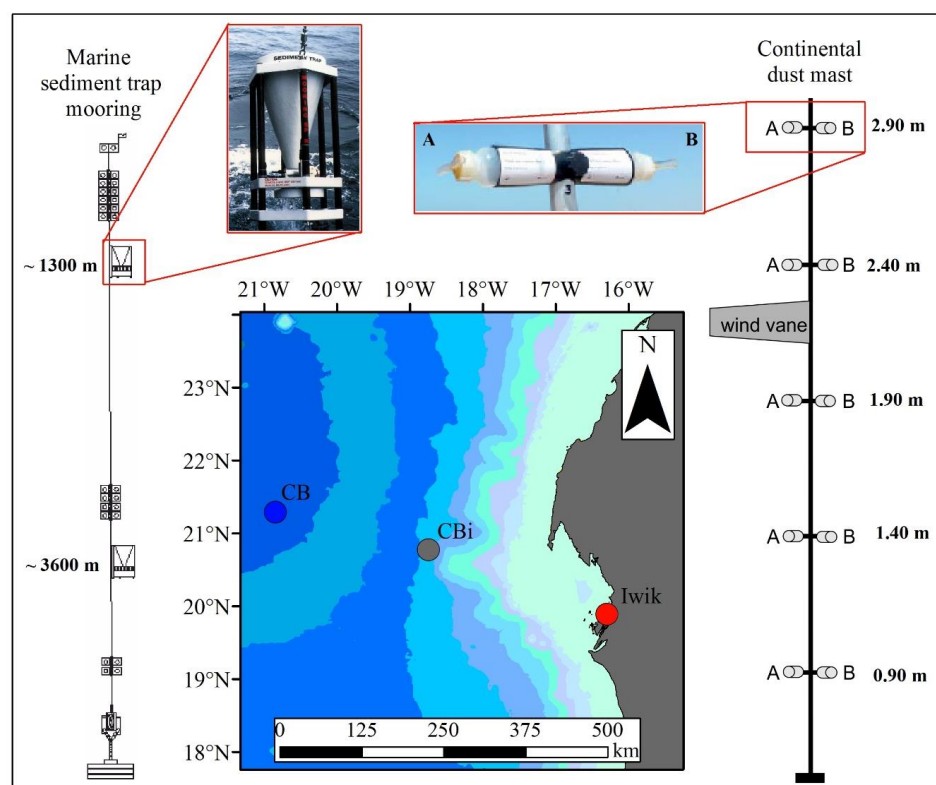

**Figure 2: The marine sediment trap moorings CB and CBi offshore Cape Blanc and the dust masts near Iwik, Mauritania. On the left, a sketch of the sediment trap mooring (sketch of CB 24 copied from Fischer et al. (2013)) together with a photograph of the trap (downloaded from www.kum-kiel.de) is displayed. On the right, a sketch of the dust mast together with a photograph of the MWAC sampling bottles is depicted.**

The sampling intervals were synchronized between the two sites. The intervals ranged from 9.5 days to 21.5 days (Table 1). Deployment and recovery of the sediment-trap samples was performed during the Research Vessel Poseidon expeditions POS445 (Fischer et al., 2013), POS464 (Fischer et al., 2014) and POS481 (Fischer et al., 2015a) (Table 1). The working steps related to the trap deployment and treatment are described in Fischer and Wefer (1991). In order to prevent outflow of water from the cups during sampling, each sampling cup was filled



with 20 ml of filtered (<0.2 µm) seawater with a salinity of 40 ‰. To produce seawater with a salinity of 40 ‰,
100 g NaCl suprapur was added to 1 l of filtered seawater. Microbial and zooplankton activity was inhibited inside
the trap samples by adding 1 ml of a saturated solution of the biocide $HgCl_2$ per 100 ml of seawater. After recovery,
swimmers <1 cm were removed from the samples by sieving each sample through a 1 mm mesh. A McLane rotary
liquid splitter was used to split the <1 mm fraction of each sample into five equal aliquots.
The samples of two sediment-trap deployments during 2013-2015 of the sediment trap mooring stations CB and
CBi were chosen for grain-size analyses (Table 1). The upper traps sampled at an average water depth of ~ 1300
m and the lower trap sampled at a water depth of ~3600 m (Table 1). Dust which settles at the ocean surface is
advected by ocean currents during settling in the water column. As a result, particles that settle in an area of ~ 40
x 40 $km^2$ in the ocean surface above the traps may be collected in a water depth of ~ 1300 m (Friese et al., 2016).
Two winter and two summer samples were chosen for X-ray Diffraction (XRD) measurements (Table 2).
**Table 1: Specifications of the sediment trap samples collected during 2013-2015 chosen for flux and grain-size analysis.**

| Trap series | Trap type | Sampling period | Cruise deployment | Cruise recovery | Position | Trap depth [m] | Water depth [m] | No. of samples | Sampling intervals |
|---|---|---|---|---|---|---|---|---|---|
| **CBi 11 upper** (GeoB 18006-2) | SMT 243 | 29.01.2013 – 25.03.2014 | Pos445 | Pos464 | 20°46.4' N 18°44.4' W | 1406 | 2800 | 18 | 17x21d, 1 x20d |
| **CBi 12 upper** (GeoB 19402-01) | SMT 234 NE | 14.02.2014 - 23.02.2015 | Pos464 | Pos481 | 20°46.4' N 18°44.5' W | 1356 | 2750 | 20 | 1x12.5 d, 19x19.5 |
| **CB 24 upper** (GeoB 18001-1) | SMT 234 NE | 24.01.2013 - 05.02.2014 | Pos445 | Pos464 | 21°16.9' N 20°50.6' W | 1214 | 4160 | 18 | 1x26 d, 16x21 d, 1x15 d |
| **CB 25 lower** (GeoB 19401-1) | SMT 234 NE | 07.02.2014 – 21.02.2015 | Pos464 | Pos481 | 21°17.8' N 20°47.8' W | 3622 | 4160 | 20 | 19x19.5 d, 1x9.5 d |


**Table 2: Sediment trap and MWAC samples chosen for mineralogical investigation.**

| Sample | Sampling period | Mast | Bottle | Elevation/water depth [m] | Sampling interval |
|---|---|---|---|---|---|
| **CBi 11 upper # 8** | 25.06.-16.07.13 | - | - | 1406 | 21d |
| **CBi 12 upper # 2** | 26.02.-18.03.14 | - | - | 1356 | 20d |
| **CBi 12 upper # 10** | 01.08.-21.08.14 | - | - | 1356 | 20d |
| **CBi 12 upper # 17** | 16.12.-04.01.15 | - | - | 1356 | 19d |
| **Iwik 13-7-2-3B** | 24.06.-15.07.13 | 2 | B | 1.90 | 21d |
| **Iwik 14-8-2-5B** | 15.08.-15.09.14 | 2 | B | 2.90 | 31d |
| **Iwik 14-12-1-4A** | 15.12.14-18.01.15 | 1 | A | 2.40 | 34d |
| **Iwik 14-2-2-5B** | 15.02.-15.03.14 | 2 | B | 2.90 | 28d |




**2.2 Modified Wilson and Cooke (MWAC) samplers**

Saharan dust was collected on land near Iwik, Mauritania, with a passive dust sampler consisting of two masts (1 and 2) with two sets of five air sampling bottles each (A and B, Fig. 2). The dust sampling bottles are referred to as modified Wilson and Cooke (MWAC) samplers (Wilson and Cooke, 1980;Mendez et al., 2011) and consist of a closed Polyethylene bottle through which the wind can pass via two glass tubes of 8 mm openings. Thus, a big difference between the traps and the MWAC collectors is the much smaller collection area of the MWAC collectors with 44 mm$^2$. The MWAC dust sampler was chosen because it is one of the most common (Zobeck et al., 2003) and most efficient dust samplers (Goossens and Offer, 2000). The sampling bottles were mounted horizontally at five different heights. The masts were aligned to the ambient wind direction via a wind vane (Fig. 2).

The samples collected in 2013-2015 were chosen for subsequent flux and grain-size analyses (Table 3). Saltating dust particles may be collected in the lower sampling bottles at 90 cm. However, the aim was to analyse dust transported in suspension to enable a better comparison between the continental and marine sites. Therefore, the highest sampling bottles attached to the mast at 2.90 m height were used for microscope, flux and grain-size analysis (Table 2). One series of bottles (series B2) of mast 2 were analysed with the microscope. The other three replicate samples (bottles A1 and B1 of mast 1, bottles A2 of mast 2) were analysed for flux and grain-size analysis. Out of the three replicate samples, the sample with the highest mass was chosen for the interpretation of the flux and grain-size data because this bottle was assumed to have sampled most efficiently. Three samples mounted at a height of 2.40 m of mast 2 were chosen to test the effect of the chemical pre-treatments that we do to isolate the terrigenous fraction from marine sediments on the resulting grain-size distributions (Fig. 2). Two winter and two summer samples that contained enough material were chosen for XRD measurements (Table 2).

Furthermore, dust was sampled with a MWAC dust sampler mounted on the mast of buoy Carmen, at about 2 m above the sea surface (Stuut et al., 2015). A wind vane was attached to the mast which aligned the sampler to the ambient wind direction. This MWAC dust sample was also analysed for grain-size distribution.

**Table 3: Specifications of the MWAC samples collected during 2013-2015 chosen for flux and grain-size analysis.**

| Dust collector series | Trap type | Sampling period | Position | Height [m] | No. of samples | Sampling intervals |
|---|---|---|---|---|---|---|
| Iwik 13 | MWAC | 27.01.2013 – 20.01.2014 | 19°53.1' N 16° 17.6' W | 2.90 | 11 | 19 d, 28 d. 32 d, 29 d, 40 d, 21 d, 31 d, 61 d, 31 d, 31 d, 35 d |
| Iwik 14 | MWAC | 20.01.2014 - 18.01.2015 | 19°53.1' N 16° 17.6' W | 2.90 | 13 | 26 d,. 28 d, 31 d, 30 d, 31 d, 30 d, 31 d, 31 d, 30 d, 32 d, 29 d, 34 d |
| CB-MWAC | MWAC | 23.08.2014 – 16.11.2015 | 21°15.8' N 20°56.1' W | 2.00 | 1 | 450 d |

**2.3 Microscopy**

The MWAC samples chosen for microscopic investigation were analysed with a Leica M165 C microscope. Microscope pictures were taken using a Leica DFC420 camera attached to the microscope. The software Leica application suite 3.8 was used for taking the pictures.





**2.4 Dust and lithogenic fluxes**


1/5 splits of the sediment trap samples were analysed for dust fluxes and the bulk components following the method
presented in Fischer and Wefer (1991). The lithogenic flux [mgm$^{-2}$d$^{-1}$] was estimated according to Eq. (1):
$$lithogenic\ material = dust = total\ mass - carbonate - opal - 2\ x\ Corg \qquad (1)$$
Organic carbon was measured after the removal of carbonate with 2N HCl using a CHN-Analyser (HERAEUS).
Total carbon was estimated by combustion without pre-treatment. Carbonate was determined according to Eq. (2):
$$carbonate = total\ carbon - organic\ carbon \qquad (2)$$
Biogenic opal was determined with a sequential leaching technique (Müller and Schneider, 1993).
The MWAC samples chosen for dust flux analyses were weighed on a Mettler-Toledo AT261 Delta Range balance
with a precision of 0.0001 g. Mean atmospheric dust concentrations were estimated as Eq. (3):
$$DL = \frac{MAR}{(v*A)} * \frac{1}{\eta} \qquad (3)$$
Where DL is the mean dust concentration [μgm$^{-3}$], MAR is the mass accumulation rate [μgs$^{-1}$], v is the mean wind
speed per sampling month [ms$^{-1}$], A is the cross-sectional area of the inlet tube of the MWAC sampler [m$^2$] and η
is the estimated sampling efficiency of MWAC bottles. A sampling efficiency of 90 % was assumed based on an
efficiency study of Goossens and Offer (2000). Mean horizontal dust fluxes were calculated according to Eq. (4):
$$F_h = \frac{MAR}{A} * \frac{1}{\eta} \qquad (4)$$
where F$_h$ is the horizontal dust flux [mgm$^{-2}$d$^{-1}$], MAR is the mass accumulation rate [mgd$^{-1}$], A is the cross-
sectional area of the inlet tube of the MWAC sampler [m$^2$] and η is the estimated sampling efficiency of MWAC
bottles.

**2.5 Particle size**


A 1/25 split of the marine sediment trap samples was analysed for particle size of the terrigenous fraction. The
samples were pre-treated before measurement in order to isolate this fraction (see also Filipsson et al. (2011);Friese
et al. (2016), Meyer et al. (2013) and Stuut (2001) for methodology) with the following steps: (1) removal of
organic matter: Addition of 10 ml of H$_2$O$_2$ (35%) to the sediment sample and subsequent boiling until the reaction
stops, (2) removal of calcium carbonate: Addition of 10 ml HCl (10%) to the sediment sample and subsequent
boiling for exactly 1 minute and (3) removal of biogenic silica: Adding 6 g of NaOH pellets to the sediment sample
and subsequent boiling for 10 minutes. Before particle-size analysis, 10 drops of Na$_4$P$_2$O$_7$*10H$_2$O were added to
each sample to assure the full disaggregation of the particles. The pre-treatment of the MWAC samples differed
from the pre-treatment of the sediment trap samples as, obviously, these samples did not contain any biogenic
material originating from marine plankton. Further, the disaggregation of particles needed to be kept at minimum
to allow for the study of dust transport processes, the so-called 'minimally dispersed' aeolian fraction (McTainsh
et al., 1997). Therefore, the MWAC samples were solely pre-treated with three drops of Na$_4$P$_2$O$_7$*10H$_2$O before
analysis. The marine sediment-trap samples as well as the MWAC samples were analysed with the laser particle
sizer Beckmann Coulter LS13320 at NIOZ using a Micro Liquid Module (MLM). This instrument allows quick,





accurate, and precise data acquisition of large size intervals (Bloemsma et al., 2012). An analytical error of ± 1.26
µm (± 4.00 %) was considered for the measurements (Friese et al., 2016).
To investigate the comparability of the MWAC samples with the oceanic sediment-trap samples, the particle-size
distribution of the MWAC sample attached to buoy Carmen was compared to the averaged particle-size
distributions of the upper and lower trap series at site CB (Fig. 3a). The grain-size distribution of the MWAC
sample was comparable to both sediment trap time series even though the sampling time period was different. To
ensure that the pre-treatment steps of the traps did not influence the terrigenous fraction itself, tests were made in
which the on-land MWAC samples were exposed to the same pre-treatment steps as the marine samples (Fig. 3b).
One spring sample has been measured with and without a chemical pre-treatment. Two fall dust samples were
obtained from the same height and mast and sampling interval, however from different bottles (A and B) and were
measured with and without pre-treatment. The figure indicates that a pre-treatment of the Iwik dust samples did
not alter the particle distributions of the samples significantly. Further, the particle-size distribution of dust sampled
with different bottles is comparable.

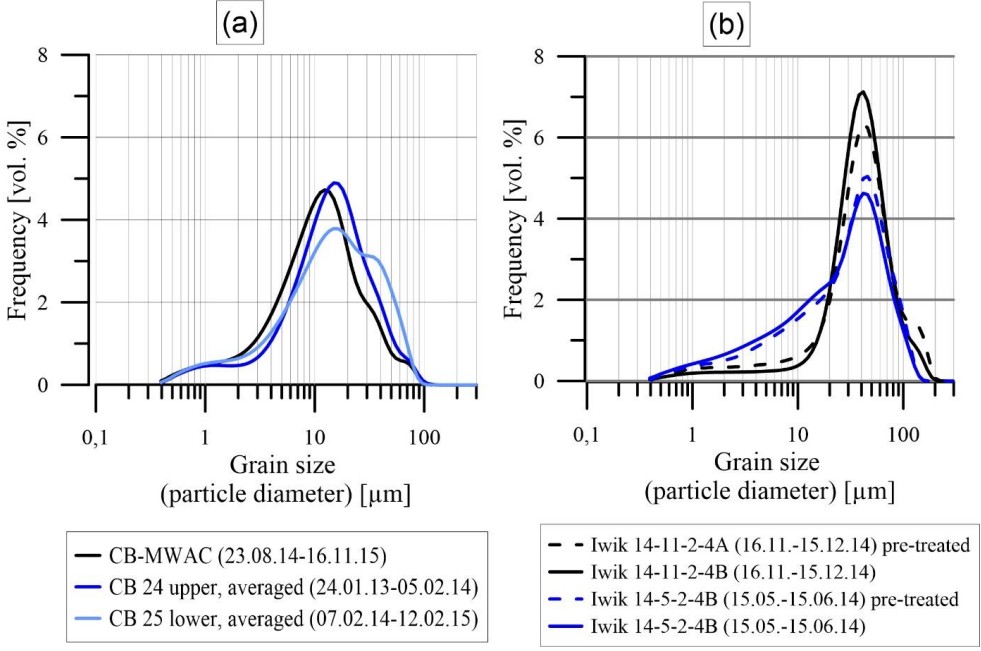


**Figure 3: (a) Grain-size distributions for the station CB: Dust sampled with the MWAC sampler 2 m above sea level,**
**with the upper sediment trap at 1214 mbsl and the lower trap at 3622 mbsl. (b) Grain-size distributions of samples of**
**the Iwik 14 time series which have been pre-treated with HCl, H$_2$O$_2$ and NaOH (dotted lines) and without pre-**
**treatment (lines).**
**2.5 Mineral assemblages**
Two winter and two summer samples of the MWAC dust collector and the sediment-trap series CBi were chosen
for XRD analysis (Table 3). X-Ray Diffraction pattern analyses were carried out in the laboratory of the research





group Crystallography (University of Bremen, Central Laboratory for Crystallography and Applied Material
Sciences, ZEKAM, Dept. of Geosciences).
Due to the small amount of material in the available dust samples (< 100 mg), the preparation for the measurement
was done by pipetting a demi-water-sample mixture on glass slides. A thorough preparation commonly increases
reproducibility of the results, however, the standard deviation given by Moore and Reynolds (1989) of ±5% can
be considered as a general guideline for mineral groups with >20% clay fraction. In addition, the determination of
well-crystallized minerals like quartz, calcite or aragonite can be done with better standard deviations (Tucker and
Tucker, 1988;Vogt et al., 2002). The X- Ray Diffraction was measured on a Philips X'Pert Pro multipurpose
diffractometer equipped with a Cu-tube ($k_\alpha$ 1.541, 45 kV, 40 mA), a fixed divergence slit of ¼°, a secondary Ni-
Filter and the X'Celerator detector system. The measurements were carried out as a continuous scan from 3 – 85°
2θ, with a calculated step size of 0.016° 2θ (calculated time per step was 100 seconds). Mineral identification was
accomplished using the Philips software X'Pert HighScore™, which, besides the mineral identification, can give
a semi-quantitative value for each identified mineral on the basis of Relative Intensity Ratio (R.I.R.)-values. The
R.I.R.-values are calculated as the ratio of the intensity of the most intense reflex of a specific mineral phase to the
intensity of the most intense reflex of pure corundum (I/Ic) referring to the "matrix-flushing method" after Chung
(1974). Unfortunately R.I.R. values are sparse for clay minerals and long chain organic materials hampered the
quantification of our samples.
**2.6 Meteorological data**
The obtained flux and size data were compared to near-by meteorological data (wind speed, wind direction and
precipitation).
Wind direction, wind speed and precipitation data with a 20 minute resolution were gathered for the sampling site
CB (21°17' N – 21°12' N, 20°56' W - 20°54' W) during the buoy Carmen deployments from November 2013 to
September 2015 with a Vaisala WXT520 meteorology sensor. The size of the dataset was reduced by calculating
four hour averages. Moreover, wind direction and wind-speed data with a resolution of five minutes to one hour
were gathered during sampling at site Iwik (19°53.1' N, 16° 17.6' W) from January 2013 to January 2015 with a
Davis 6250 Vantage Vue meteorology sensor. The size of the dataset was reduced by calculating one-hour
averages. Further hourly precipitation data were gathered from the station Arkeiss (20° 7' N, -16° 15' W) from
December 2013 to March 2015 with another Davis 6250 Vantage Vue meteorology sensor. Continental hourly
wind direction and wind-speed data was acquired for the Nouadhibou meteorological station (20° 55' N, 17° 1'
W) online from the Cedar Lake Ventures website (https://weatherspark.com).
Local daily precipitation data (TRMM 3B42 dataset, 0.25° spatial resolution) were derived from the Giovanni
online data system, developed and maintained by the NASA GES DISC (http://gdata1.sci.gsfc.nasa.gov). Daily
precipitation data were downloaded as area-averages around CBi (20° 58' N - 20° 34' N, 18° 56 W - 18° 32' W),
Iwik (19° 41' N - 20° 5'N, 16° 29' W - 16° 05' W), CB/Carmen (21° 05' N - 21° 29' N, 21° 02' W - 20° 38' W) and
Arkeiss (20° 19' N - 19° 55' N, 16° 28' W - 16° 04' W) according to the assumed catchment area of the upper trap
(~ 40 x 40 km²).






**2.7 Mapping with ArcMap**
The mapping software ArcMap version 10.3.1 was used to analyze the source regions of the dust samples
investigated for mineralogical composition. A map was created with four-day back-trajectories for days with a
dust-storm event as depicted on satellite images. In addition, the African surface lithology was included in the map
and soils rich in the minerals calcite, kaolinite and chlorite were marked.
Satellite quasi-true colour RGB images (MODIS dataset) were retrieved from the NASA Ocean Biology
Distributed Active Archive Centre (*OB.DAAC*), Goddard Space Flight Centre, Greenbelt MD, on their website
(http://oceancolor.gsfc.nasa.gov).
Four-day back trajectories (at altitudes of 10, 3000, 4500 and 5500m) were calculated ending at the dust collector
site Iwik (19°52' N, 16°17' W) and at the proximal marine trap site CBi (20°46',18°44' W) using the Hybrid
Single Particle Langrangian Integrated Trajectory (HYSPLIT) model (Stein et al., 2015) and the reanalysis dataset
(2.5° spatial resolution) on the NOAA website at http://ready.arl.noaa.gov.
An ArcGIS layer file of the African surface lithology
(new_af_lithology_w_glbcvr_waterbdy_90m_dd84_final.lyr) was downloaded from the U.S. Geological survey
(USGS) website: http://rmgsc.cr.usgs.gov/outgoing/ecosystems/AfricaData.
An ArcGIS shape file of the African soils (DSMW.shp) was downloaded from the website of the food and
agriculture organization of the United Nations (FAO) at
http://www.fao.org/geonetwork/srv/en/metadata.show?id=14116. The mean percentage of calcite, chlorite and
kaolinite in the clay fraction of Saharan soils in general and for each soil type is given by Journet et al. (2014).
The average percentages of calcite, chlorite and kaolinite in the clay fraction of Saharan soils are 8.9 %, 4.1 % and
29 %, respectively (Journet et al., 2014). Soils with larger percentages of calcite, chlorite or kaolinite in the clay
fraction than the average percentages were marked in the ArcGIS map.













## 3. Results

### 3.1 Meteorology

In Fig. 4 the meteorological data of the sites Carmen/CB, CBi, Iwik, Arkeiss and Nouadhibou during 2013 to 2014 are presented (see Fig 4a for location of the sites). The rainfall frequency is given in Fig. 4b for each site. The number of rainfall events were calculated regarding the TRMM stations for precipitation rates >1 mmd$^{-1}$ because smaller precipitation amounts which were detected by the satellite may not actually reach the ground.  Regarding the ground stations Carmen and Arkeiss, a threshold of >0.2 mmd$^{-1}$ was used in order to exclude events which may be related to anomalously high moisture instead of rainfall.

According to the TRMM satellite product the annual precipitation frequency was larger on the shoreline (station Arkeiss and Iwik) than offshore (station CBi and Carmen) (Fig. 4b). This may be explained by a decrease in atmospheric water vapor content due to precipitation when the winds move westward. Moreover, the TRMM satellite product indicated larger rainfall frequencies during the summer season compared to the winter season regarding the oceanic stations Carmen, CBi, Iwik and Arkeiss. Larger summer rainfall frequencies can be explained by the summer northward shift of the ITCZ to ~ 21° N resulting in more frequent moist convection and rainfall in the study area.

The annual rainfall frequency at the site Arkeiss and the summer rainfall frequencies at the sites Arkeiss and Carmen compare quite well between the sensors and the TRMM observations. However, the spatial and seasonal trends observed by the TRMM data were not supported by the sensor on buoy Carmen and by the ground station in Arkeiss. The larger annual and winter rainfall frequency recorded with the sensor on buoy Carmen may be related to water emission from the ocean surface during time periods with strong surface winds. Further, disagreements between the ground stations and the TRMM stations maybe caused by the local signal recorded by the respective rain sensor. A larger number of rain sensors would most likely improve the comparability to the TRMM data.






**Figure 4: Meteorological data (a) map showing the study sites CB, CBi and Iwik and the meteorological station in**

**Nouadhibou under investigation (b) precipitation at the study sites CB, CBi and Iwik (c) wind direction and speed at**

**the study sites CB and Iwik and at the meteorological station in Nouadhibou.**

The wind direction and speed for the ground stations Carmen, Nouadhibou and Iwik are displayed in Fig. 4c. The
annual average surface wind velocity was maximum offshore at buoy site Carmen/CB with ~ 8 m/s. The buoy





recorded a larger average wind velocity during winter than during summer, which is consistent with this season
being dominated by the Trades. On the shoreline, the average wind velocity was slightly larger during summer
than during winter. The predominant annual wind direction was NE at site Carmen and Iwik, while predominant
NW winds were recorded for the site Nouadhibou. The wind direction changed from predominant NE during
winter to predominant NNE direction during summer at site Carmen. A similar, but less pronounced seasonal trend
can be observed for the continental site Iwik. In Nouadhibou, the predominant winter wind direction is NNW
switching to a predominant NW wind direction during summer. Obviously, with winds originating from the open
ocean, not a lot of dust is anticipated. Therefore, we interpret these wind directions as being very local and caused
by the shape of the peninsula of Cape Blanc.
**3. 2 Microscope findings of the dust samples from Iwik**
In Fig. 5 the results of the microscopy investigation of the Iwik 2013 time series are presented. In general, the
majority of the particles consisted of angular and moderately spherical quartz grains with a diameter of ~ 50 μm
(Fig. 5a,b). A small percentage of large platy minerals with a diameter of ~ 200 μm were found in all samples (Fig.
5b). Large quartz grains with a diameter of ~ 150 to 200 μm were detected in 45 % of the samples. An anomalously
high percentage of sub-angular and moderately spherical quartz grains with an average diameter of ~ 200 μm was
observed in one summer sample (Fig. 5c). Aggregated grains occurred in all samples. However, the percentage
and size of the aggregates as well as the size of the aggregated grains differed from sample to sample. Usually, the
size of the aggregated grains was ~ 50 μm (Fig. 5a). Two samples were characterized by aggregates composed of
particles with a smaller size of ~ 20 μm (Fig. 5d).

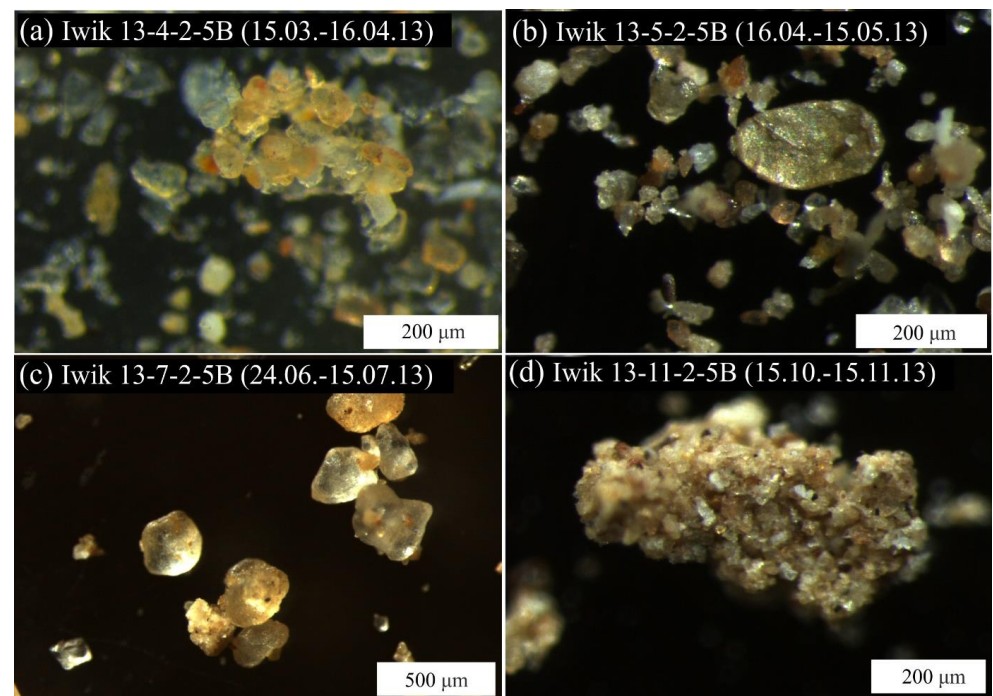




**Figure 5: Microscopic photographs of selected dust samples from the Iwik 2013 time series. (a) Spring dust sample with**
**a ~ 250 x 150 µm aggregate, (b) spring dust sample with a ~ 200 x 100 µm  mica chip, (c) summer dust sample with ~**
**200 x 200 µm quartz grains, (d) fall dust sample with a ~ 600 x 250 µm aggregate.**
**3.3 Dust fluxes and size on land and in the ocean**
In Table 4 the average dust fluxes are given for the sampling sites Iwik, CBi and CB. The dust concentrations at
site Iwik were determined based on the measured wind speed of the meteorological sensor attached to the sampling
mast. For four samples no wind data were available due to a failure of the instrument. For these samples a wind
velocity was assumed based on the seasonal averages calculated from the available wind data of the meteorology
sensor in Iwik (Fig. 4c). The annual average horizontal dust fluxes at site Iwik were of the same order of magnitude
during 2013 and 2014. The $PM_{10}$ concentration was calculated in order to enable a comparison to other study sites
where only dust particles smaller than 10 µm were sampled. The annual average dust fluxes decreased from the
on-land site Iwik towards the proximal site CBi and the distal site CB. The dust fluxes were about 1000 times
smaller at the oceanic sites compared to the continental site. A stronger decrease in the fluxes was observed from
site Iwik to CB during summer compared to winter. The variation in the seasonal average dust fluxes were well
comparable between the continental and oceanic site CBi. A seasonal trend in the dust fluxes could not be observed
for the ocean sites CBi and CB. However, a seasonal trend was observed for site Iwik when taking into account
the spring and fall samples. The average dust concentration was maximum during spring plus winter 2013 and
2014 with 393 µgm$^{-3}$ and 341 µgm$^{-3}$, respectively, and minimum in fall 2013 and 2014 with 48 and 68 µgm$^{-3}$,
respectively. The dust fluxes generally decreased with collection height in the mast between 90 and 290 cm (not
shown).
**Table 4: Seasonal and annual average dust fluxes and average modal grain size, mean/mode ratio and standard**
**deviation of the grain-size distributions from Iwik 13-14, CBi 11-12 upper and CB 24 upper time-series.**

| Series | Year | Winter | Summer | Annual |
|---|---|---|---|---|
| *Average dust fluxes [mg.m⁻².d⁻¹] (dust concentration  [µg.m⁻³])* | | | | |
| Iwik 13 | 2013 | 10000 (30) | 113000 (268) | 95000 (214) |
| CBi 11 upper | 2013 | 106 | 168 | 99 |
| CB 24 upper | 2013 | 53 | 44 | 45 |
| Iwik14 | 2014 | 208000 (603) | 55000 (127) | 102000 (275) |
| CBi 11+12 upper | 2014 | 98 | 20 | 47 |
| *Average modal grain size [µm]* | | | | |
| Iwik 13 | 2013 | 44 | 49 | 48 |
| CBi 11 upper | 2013 | 27 | 39 | 29 |
| CB 24 upper | 2013 | 16 | 17 | 16 |
| Iwik 14 | 2014 | 45 | 49 | 48 |
| CBi  11+12 upper | 2014 | 34 | 44 | 33 |
| *Average mean/mode ratio [µm]* | | | | |
| Iwik 13 | 2013 | 0.7 | 0.6 | 0.6 |
| CBi 11 upper | 2013 | 0.5 | 0.3 | 0.5 |
| CB 24 upper | 2013 | 0.7 | 0.8 | 0.7 |
| Iwik14 | 2014 | 0.6 | 0.4 | 0.6 |
| CBi 11+12 upper | 2014 | 0.5 | 0.3 | 0.5 |
| *Average standard deviation [µm]* | | | | |
| Iwik 13 | 2013 | 2.8 | 3.1 | 3.0 |
| CBi 11 upper | 2013 | 3.0 | 3.3 | 3.1 |
| CB 24 upper | 2013 | 2.7 | 2.6 | 2.6 |
| Iwik 14 | 2014 | 2.8 | 3.5 | 3.1 |
| CBi 11+12 upper | 2014 | 3.1 | 3.3 | 3.0 |





The statistical values of the measured grain-size distributions for the stations CB, CBi and Iwik are given in Table
4. In addition, the measured grain-size distributions for the time series of the stations CB, CBi and Iwik are
displayed in Fig. 6. In Fig. 6a the average grain-size distribution for the samples of each of the three stations for
the year 2013 are given. The maximum measured particle size decreased from ~223 μm on land at site Iwik to
~169 μm at the proximal site CBi and ~140 μm at the distal site CB (Fig. 6a). In addition, the average modal grain
size decreased from ~48 μm at site Iwik to 16 μm at site CB (Table 4). Bimodal grain-size distributions were
encountered for 23 % of the CBi 11-12 samples, 13 % of the Iwik 13-14 samples, and none of the CB 24 samples.
The bimodal distributions of the Iwik 13-14 time series were characterized by an additional fine mode peaking at
~16 μm besides the more pronounced and variable coarse mode peaking at ~42 to 55 μm. Two of the Iwik dust
samples characterized by a fine grain-size peak were collected during summer and fall respectively. The sorting
of the CB samples was better than the sorting of the Iwik and CBi time series as indicated by the average geometric
standard deviations of 2.6 μm for CB and 3.1 μm for both Iwik and CBi (Table 4). The lowest average mean/mode
ratio was recorded for the CBi time-series with ~0.5 due to the weak sorting of the samples (Table 4).
In Fig. 6b-c the measured grain-size distributions for winter and summer samples are displayed. The averaged
modal grain size for the summer samples was coarser grained compared to the winter samples of the respective
grain-size time series (Table 4). The seasonality in modal grain size was largest for the CBi 11 upper trap series
of the year 2013 with a difference of ~12 μm (Table 4). The average standard deviation was larger and the average
mean/mode ratio was smaller in the summer samples compared to the winter samples regarding the sites Iwik and
CBi (Table 4). In other words: the summer samples of sites CBi and Iwik were less well sorted (Fig. 6b and c).
This seasonal trend was not observed in the CB 24 upper samples which were generally well sorted. (Table 4).

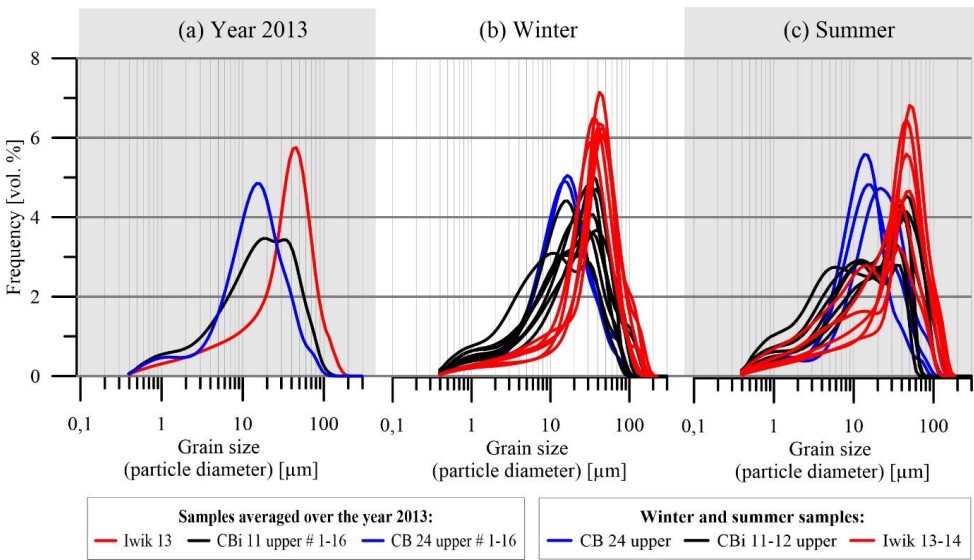


**Figure 6: Grain-size distributions of the stations Iwik, CBi and CB (a) averaged for the samples of the year 2013 (b)**
**winter samples (c) summer samples.**



In Fig. 7a –c the results of the correlation between the characteristics of the dust sampled on land and the local
meteorological data are presented. In Fig. 7a the particle sizes were correlated to the surface wind speed data (N =
13 samples). A correlation above a coefficient of determination ($R^2$) of 0.3 was considered significant at the 95 %
confidence level for two-tailed probabilities. The modal particle size of the Iwik samples showed a positive linear
correlation with the daily wind speed events with $R^2 = 0.5$, which is significant at the 99.31 % confidence level. A
better positive linear correlation was obtained when excluding the spring sample resulting in $R^2 = 0.7$ which is
significant at the 99.96 % confidence level.
In Fig. 7b the dust fluxes were correlated to the surface wind-speed data (N = 10 samples). A correlation above $R^2$
= 0.4 was considered significant at the 95 % confidence level for two-tailed probabilities. The horizontal dust flux
of the Iwik samples correlated positively to the daily wind speed events during the sampling interval with $R^2 = 0.7$
which is significant at the 99.75 % confidence level. Moreover, a significant linear correlation with $R^2 = 0.6$ was
observed at the 99.15 % confidence level between the dust fluxes and the mean wind strengths during the sampling
intervals (not shown).
In Fig.7c the particle size of the Iwik summer samples was correlated to the local TRMM precipitation data (N= 6
samples). In this case a correlation above $R^2 = 0.7$ was considered significant at the 95 % confidence level for two-
tailed probabilities. A good linear negative correlation with $R^2 = 0.9$ was observed which is significant at the 99.78
% confidence level.

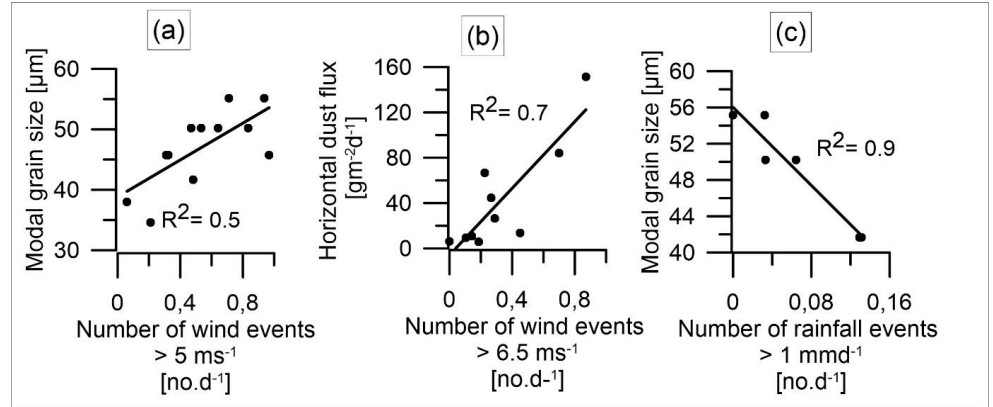


**Figure 7: Correlation between the observed local surface wind speed at site Iwik and the measured (a) modal grain size**
**and (b) flux. (c) Correlation between the observed local precipitation at site Iwik (TRMM data) and the modal grain**
**size of the summer samples.**
**3.4 Mineral assemblage of dust sampled on land and in the ocean**
In Table 5 the mineralogical composition averaged over all eight samples, averaged over the four Iwik samples
and the four CBi samples is given. All dust samples contained the minerals quartz and mica. Further minerals that
occurred with significant quantities but which were not present in all dust samples were feldspar, amphibole,
zeolite, chlorite and palygorskite. Calcite, dolomite, gibbsite, kaolinite, smectite, sepiolite, fluellite, anhydrite,
rutile and serpentine occurred only in some samples resulting in a low average abundance ≤ 1%. However, we
argue that these minerals can be used as dust source indicators because of (1) the characteristic distribution of





gibbsite, kaolinite, smectite and sepiolite in North Africa according to different weathering regimes (Biscaye,
1964) and (2) the characteristic occurrence of fluellite, anhydrite, rutile and serpentine according to outcropping
rock type (Deer et al., 1992). Further minerals that occur in low abundances (≤ 3%) were summarized as 'other
minerals' and will not be discussed in the manuscript. While the continental samples were dominated by quartz
and feldspar, the marine samples were dominated by mica, followed by quartz and feldspar.
**Table 5: Results of the mineralogical investigation: Mineral assemblage averaged over all samples (Total), the Iwik**
**samples (Iwik) and the CBi samples (CBi).**

| * | Qz [%] | Fsp [%] | Mi [%] | Amf [%] | Pal [%] | Chl [%] | Cc [%] | Dol [%] | Gib [%] | Zeo [%] | Kao [%] | Sme [%] | Se [%] | Rut [%] | Serp [%] | Ga [%] | Anh [%] | Flu [%] |
|---|---|---|---|---|---|---|---|---|---|---|---|---|---|---|---|---|---|---|
| **Total** | 25.1 | 21.5 | 25.5 | 5.1 | 3.4 | 4.4 | 0.6 | 0.1 | 1.0 | 3.8 | 0.9 | 0.4 | 1.1 | 0.5 | 0.3 | 0.1 | 0.1 | 1.1 |
| **Iwik** | 33.3 | 30.8 | 18.0 | 5.0 | 3.3 | 1.8 | 1.3 | 0.3 | 2.0 | 0.0 | 0.0 | 0.0 | 0.0 | 0.8 | 0.5 | 0.0 | 0.0 | 0.0 |
| **CBi** | 17.0 | 12.3 | 33.0 | 5.3 | 3.5 | 7.0 | 0.0 | 0.0 | 0.0 | 7.5 | 1.8 | 0.8 | 2.3 | 0.3 | 0.0 | 0.3 | 0.3 | 2.3 |
| *Qz = quartz, Fsp = feldspar, Mi = mica, Amf = amphibole, Pal = palygorsite, Chl = chlorite, Cc = calcite, Dol = dolomite, Gib = gibbsite, Zeo = zeolite, Kao = kaolinite, Sme = smectite, Se = sepiolite, Rut = rutile, Serp = serpentine, Ga = garnet, Anh = anhydrite, Flu = fluellite | | | | | | | | | | | | | | | | | | |


In Fig. 8a-c the results of the mineralogical investigation of the eight chosen dust samples are presented. Figure 8a
depicts again the average composition of the samples per sampling site (N=4). The minerals zeolite, anhydrite,
garnet, sepiolite, fluellite, kaolinite and smectite were only found in the marine samples. Only the continental
sample of 15.08.-15.09.14 contained traces of zeolite. While gibbsite, serpentine, calcite and dolomite were
detected in the continental dust samples, these minerals were absent in all marine samples. The absence of calcite
and gibbsite may have been caused by the pre-treatment of the marine sediment-trap samples with HCl. Although
the concentration of the used acid is fairly low (10%) and the exposure time of the samples was exactly 1 minute,
we cannot exclude that carbonate minerals were dissolved. Therefore, the absence of these minerals in the marine
traps will not be discussed further.
In the following, the seasonality in the average mineralogical composition will be outlined for each site as given
in Fig. 8b,c. At site Iwik, the winter dust samples were characterized by the occurrence of chlorite, serpentine and
rutile, while the summer samples were characterized by the minerals gibbsite and dolomite. At site CBi, the winter
dust samples were characterized by the occurrence of the minerals sepiolite, fluellite, kaolinite, smectite, garnet
and anhydrite, while the summer samples were characterized by the mineral rutile. Only for the marine trap
samples an annual average chlorite/kaolinite ratio (C/K = 4) could be derived owing to the occurrence of kaolinite.





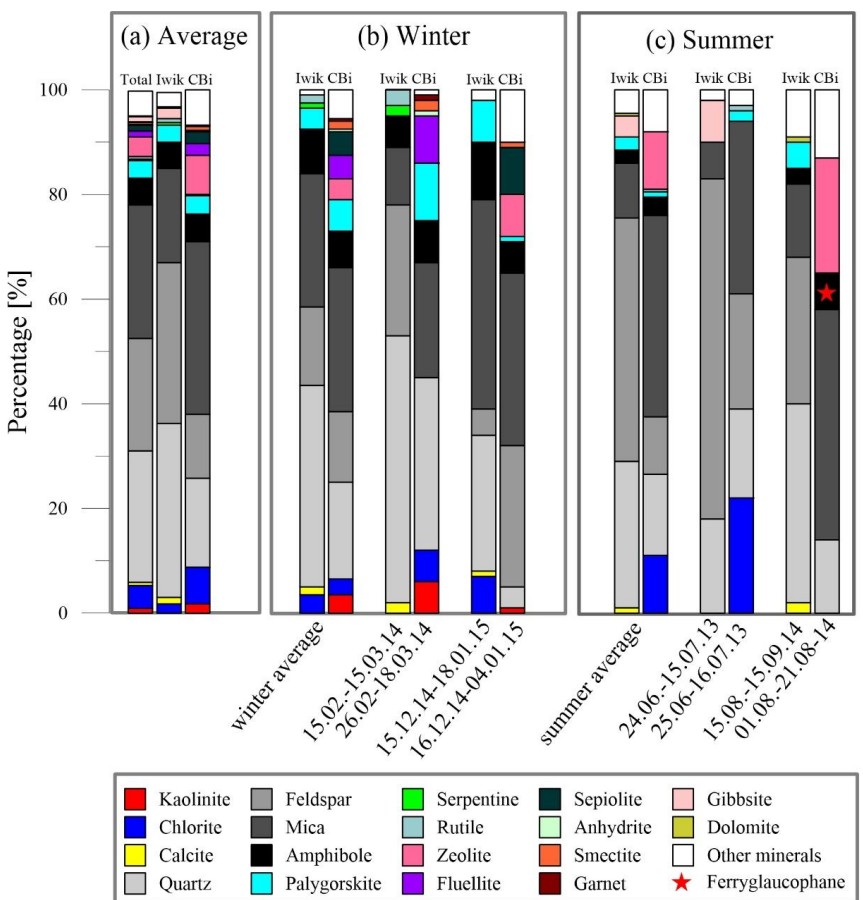

**Figure 8: Mineralogical composition (a) averaged over all samples and for sites Iwik and CBi, (b) averaged for the winter samples at sites Iwik and CBi and for each individual winter sample and (c) averaged for the summer samples at sites Iwik and CBi and for each individual summer sample. The category 'other minerals' comprises the minerals todorokite, sodalite, konicklite, guyanaite, nitratnine, urea, bernalite, akermanite, mixed-layer clay and talc.**

**3.5 Identification of dust source regions using ArcMap**

In Fig. 9-12 the results of the four day back-trajectory analysis are presented for each sample which has been analyzed for mineralogical composition. Two heights, 10 m (according to Stuut et al. (2005)) and 4500 m (according to Skonieczny et al. (2013)) were chosen to cover both low- (trades) and high-level (SAL) dust transport. Only the low-level back-trajectories were plotted for site Iwik because of the correlation of the measured dust characteristics to the low-level wind speed. Moreover, the MWAC samplers were designed to only sample dry deposition, whereas the marine sampling sites collect material settling through the water column, i.e., dust resulting from both dry- and wet deposition. The back-trajectories at 3000 and 5500 m can be found in the supplements.

Figure 9 illustrates a typical late-winter situation. During the sampling interval at least two dust storms occurred (Fig. 9c,d). Both the low-level back-trajectories ending at the continental trap site Iwik and at the oceanic trap site



CBi point to a dust source within the major **PSA 2** (Scheuvens et al., 2013). Some calcite was present in the
continental dust sample, but no chlorite nor kaolinite was detected. Therefore, the dust source was most likely
located in the nearby southwestern Reguibat Shield where sediments are rich in calcite and quartz and depleted in
chlorite and kaolinite (Fig. 9a). Dust deposited in the marine traps during the time interval was characterized by
the occurrence of chlorite and kaolinite. A small area of chlorite-rich sediments is located in the shoreline of the
Western Sahara (Fig. 9b). However, kaolinite is not present in anomalously high amounts in the proposed source
area (Fig. 9b).





**Figure 9:** Low-level (10 m) four-day back trajectories of dust events ending during the sampling interval 15.02.-15.03.14 at site Iwik and during the sampling interval 26.02.-18.03.14 at site CBi. The potential dust source areas and the mineralogy of the samples are given in the subfigures a-b. The dust-storm events occurring during the sampling interval are indicated in subfigures c-d.

Figure 10 represents a typical early-winter situation. During the sampling interval at least three dust storms occurred (Fig. 10e-g) and one lasted for several days which complicates the determination of the likely dust source





areas. All back trajectories pass through the major **PSA 2** and some point to the **PSA 1** and **PSA 3** (Scheuvens et
al., 2013). Dust sampled in the marine traps during this sampling interval did not contain any chlorite, while the
dust trapped at Iwik did. Chlorite may have been supplied to Iwik from a source area nearby the Senegal-
Mauritania Basin (Fig. 10a) or as far as the eastern Taoudeni Basin (Fig. 10b) due to the anomalously high chlorite
content of the soils in these areas. The continental sample is further characterized by the occurrence of calcite and
the absence of kaolinite which fits to the soils of the chosen source areas (Fig. 10a,b). The marine sample was
characterized by the occurrence of zeolite and absence of chlorite. Therefore, zeolite may have been derived from
the extrusive volcanic rocks of the northern Taoudeni Basin (Fig. 10c). A further source area might be the southern
shoreline of Western Sahara similar to what was observed for the sample obtained during winter 2014 (Fig. 9b,
Fig. 10d). Again, the marine winter sample contained the mineral kaolinite which cannot be explained with the
back-trajectories and the soil map.



**Figure 10: Low-level (10 m) four-day back trajectories of dust events ending during the sampling interval 15.12.14-18.01.15 at site Iwik and during the sampling interval 16.12.14-04.01.15 at site CBi. The potential source areas and the mineralogy of the samples are given in the subfigures a-c. The dust storm events occurring during the sampling interval are indicated in subfigures e-g.**

In Fig. 11 a typical early-summer situation is presented. Only one dust storm event was observed during the sampling interval (Fig. 11c). The low-level back trajectories ending at site CBi run offshore. The low-level back



trajectory ending at site Iwik passes through the major **PSA 2** and the high-level back trajectory passes through
the major **PSA 2** and **3** (Scheuvens et al., 2013). Dust sampled on land at site Iwik was characterized by the absence
of chlorite, kaolinite and calcite which fits to the soils of northern Tidra Island (Fig. 11a). In contrast, dust sampled
offshore at site CBi was characterized by chlorite and by the absence of kaolinite which fits to the chlorite rich
soils in the Mauritanides of Mauritania (Fig. 11b).





**Figure 11: High- (4500 m) and low-level (10 m) four-day back trajectories of a dust event ending during the sampling**
**interval 24.06.-15.07.13 at site Iwik and during the sampling interval 25.06.-16.07.13 at site CBi. The potential source**
**areas and the mineralogy of the samples are given in the subfigures a-b. The dust storm event is indicated in subfigure**
**c.**
In Fig. 12 a typical late-summer situation is illustrated. At least five separate dust events could be identified (Fig.
12f-j) of which one lasted for two days. The low-level back trajectories ending at site CBi run offshore. The low-
level back trajectories ending at site Iwik pass through the major **PSA 2**. The high-level back trajectories pass
through the major **PSA 2**, **PSA3** and **PSA 4**  (Scheuvens et al., 2013). Dust deposited in the continental traps was
characterized by the presence of calcite and the absence of chlorite and kaolinite. Therefore, the source area of the
dust was most likely in the Western Sahara where soils rich in calcite but poor in chlorite and kaolinite are located
(Fig. 12a,b). Dust sampled with the oceanic traps during this sampling interval was characterized by the absence
of chlorite and kaolinite and by the presence of a high percentage of zeolite (22 %) (Fig. 8c). Therefore, a possible
source area may have been extrusive volcanic rocks of the northern Taoudeni Basin (Fig. 12c) and the Fezzan
uplift (Fig. 12e). Ferryglaucophane may have been sourced by the Pharusian belt (Fig. 12d).






**Figure 12: High- (4500 m) and low-level (10 m) four-day back trajectories of dust events ending during the sampling**

**interval 15.08.-15.09.14 at site Iwik and during the sampling interval 01.08.-21.08.14 at site CBi. The potential source**
**areas and the mineralogy of the samples are given in the subfigures a-c. The dust storm events are indicated in**

**subfigures e-i.**




### 4. Discussion

### 4.1 Comparison of dust collected on land and in the ocean

#### 4.1.1 Dust concentrations

Annual average dust concentrations of ~214 (2013) and 275 μgm$^{-3}$ (2014) were estimated for the site Iwik (Table
4). These estimates were larger than what has been measured for background dust concentrations in Morocco
which were in the order of 100 μgm$^{-3}$ during spring 2006 (Kandler et al., 2009). However, in Morocco dust was
collected at a larger height of 4 m and the sampling time is much shorter leading to the monitoring of less dust
events. The horizontal dust fluxes at site Iwik correlated positively to wind speed (Fig. 7b) and decreased with
collection height (not shown). This underscores the proximity of this continental site to the dust emission source.
At the distal oceanic site CB, the annual average dust deposition flux was ~ 45 mgm$^{-2}$d$^{-1}$ (Table 4). The dust flux
was slightly larger than the average annual dust flux observed at site CB between 1988 and 2012 with ~ 30 mgm$^{-2}$d$^{-1}$
$^{2}$d$^{-1}$ (Fischer et al., 2015b). The slightly larger dust fluxes may have been caused by the anomalously high frequency
in dust storm events as observed on satellite images occurring during the studied time period (not shown). The
average horizontal fluxes at site Iwik were ~ 1000 times larger with ~ 100000 mgm$^{-2}$d$^{-1}$ (Table 4) due to the
different sampling technique. The MWAC samplers do not measure deposition fluxes but foremost dust
concentrations. Only 1% or less drops out of a moving dust cloud, hence, the horizontal dust flux is at least ~100
times higher than the dust deposition flux (Goossens, 2008). In addition, the observed general decrease in the dust
flux from the site Iwik to the sites CBi and CB can be explained via the increase in the distance to the source area.
Decreased dust deposition fluxes offshore NW Africa with increasing distance from the African coast were also
observed by Bory and Newton (2000) analysing the lithogenic fluxes in marine sediment traps. The stronger
decrease in the dust fluxes from site Iwik to CB during summer compared to winter (Table 4) may be explained in
the following. During summer, dust was additionally transported with the trades to the site Iwik, (Fig. 9-12) leading
to anomalously higher dust deposition at site Iwik compared to the oceanic sites. Further, the washout of dust
during offshore transport may have depleted the atmospheric dust cloud resulting in strongly decreased dust
deposition fluxes at site CB compared to site CBi during summer.

#### 4.1.2 Dust transport

The measured grain-size distributions for dust trapped at 2.90 m on land at site Iwik and for dust settling in the
ocean were nearly all unimodal (Fig. 6). Unimodal grain-size distributions are typical for wind-blown sediments
(Pye, 1995). Unimodal grain-size distributions were also measured for dust deposited in a vertical dust sampler in
M'Bour (Skonieczny et al., 2011), dust sampled on ship vessels (Stuut et al., 2005) and in other sediment trap
samples offshore NW Africa (Ratmeyer et al., 1999b;Friese et al., 2016;Van der Does et al., 2016a).
The measured annual average modal grain size at site Iwik was 48 μm (Table 4). The obtained average annual
modal grain size was close to the coarse mode of 44 μm observed by Gillies et al. (1996) for dust trapped at a
height of 10 m during spring in Fakarbé (Mali) which is located about 700 km southeast of Iwik. Gillies et al.
(1996) conclude that the coarse mode in the dust samples points to locally-derived dust. Based on this observation,
we argue that also the dust trapped near Iwik was most likely generally of regional instead of long-distance
provenance. The distance to the main source area may be, however, not in the direct surrounding of the dust
collector since dust sampled with MWAC samplers in the vicinity of barchan dunes of the Bodélé depression at





2.4 m height is characterized by a larger modal particle size of ~ 100 µm (Chappell et al., 2008). The annual
average modal and maximum particle size gradually decreased from the on-land site Iwik, to the proximal oceanic
site CBi and the distal oceanic site CB (Table 4, Fig. 6a). This decrease in particle size between the stations CB
and CBi was observed before and was attributed to the preferred gravitational settling of coarse particles during
dust transport (Friese et al., 2016). Moreover, many studies have confirmed a downwind fining of the terrigenous
fraction of surface sediments offshore NW Africa (Radczewski, 1939;Lange, 1975;Fütterer, 1980;Koopmann,
1981;Holz et al., 2004), and it is intuitively logical.
The three samples of the Iwik time series that were characterized by an additional small peak in the grain-size
distribution around ~ 16 µm were sampled during sampling intervals of anomalously high wind velocity. The back-
trajectories of one of these samples pointed towards a proximal and more distal dust source (Fig. 12a,b). Therefore,
it may be possible that wind velocities were high enough during the sampling interval to transport dust from more
distant sources (Fig. 12b) to the sampling site resulting in the small peak in the grain-size distributions. On the
other hand, microscopic examination prior to particle-size analyses of the Iwik samples revealed that the samples
included many aggregates (Fig. 5d). Hence, locally derived aggregates may have been sampled during periods of
high wind velocities. These aggregates may have been dispersed in the demineralized water during the
measurement of the laser resulting in the observed additional fine peak at ~16 µm. Further, precipitation was
encountered according to the TRMM data during the sampling interval of two of these three samples. Therefore,
a further explanation for the bimodal grain-size distributions may be the deposition of finer dust particles from
higher altitude of the SAL due to precipitation. Two of the three oceanic samples that were characterized by
bimodal grain-size distributions have several proposed dust source areas each (Fig. 10, 12). Thus, the sampling of
long- as well as short-travelled dust may have resulted in a bimodal grain-size distribution.
At the on-land site Iwik, a positive correlation between the modal grain sizes and wind velocities was observed
(Fig. 7a). This implied that dust was transported with the trade winds from sources of a quite constant distance
year-round. During dust storm events particles with a diameter of 40 to 50 µm may be transported ~ 100 km (Tsoar
and Pye, 1987). The proposed source areas all fall in this range except for the winter sample of 2014-2015 (Fig.
10). The winter sample was characterized by an anomalously low modal grain size of 38 µm and particles of this
size may be transported more than 100 km during dust storm events (Tsoar and Pye, 1987). Moreover, Van der
Does et al. (2016a) observed how particles up to 100 µm were transported ~ 3500 km across the Atlantic Ocean.
**4.1.3 Dust mineralogical composition**
In the dust sampled at Iwik the minerals quartz, feldspar, mica, amphibole, palygorskite, chlorite, calcite, dolomite,
gibbsite, rutile and serpentine were present (Fig. 8a). The observed occurrence of the minerals quartz, feldspar,
mica, chlorite and calcite has also been described for the bulk size fraction of soil samples and dust samples
collected in Mauritania (Schütz and Sebert, 1987). Palygorskite, mica and chlorite have also been detected by
Skonieczny et al. (2013) in the $PM_{30}$ size fraction of a three-year time series of dust deposition at M'Bour, Senegal,
more than 500km south of Iwik, Mauritania. Smectite and kaolinite, which were absent in the Iwik samples, were
the dominant minerals of the dust sampled at M'Bour (Skonieczny et al., 2013). Smectite and kaolinite are
considered as indicative for wet tropical soils and their relative abundance in soils increases southwards along the
northwest African coast (Biscaye, 1964;Lange, 1982). We argue that the mineralogical differences between the
two sites are explained by the >500 km distance between Iwik and M'Bour and the fact that the latter station is



surrounded by tropical soils. Gibbsite, rutile and serpentine have not been reported in any continental dust study
so far and thus seem to be indicative for locally-derived dust (Fig. 9a, Fig. 11a).
The dust sampled at the proximal marine site CBi contained the minerals quartz, feldspar, mica, amphibole,
palygorskite, chlorite, zeolite, kaolinite, smectite, sepiolite, rutile, garnet, anhydrite and fluellite (Fig. 8a). The first
seven of these minerals were also found in the clay and/or silt and sand fraction of Saharan dust sampled during
ship cruises parallel to the coast about 70 km off Cape Blanc (Chester et al., 1971) and perpendicular to the coast
about 80 to 180 km off Cape Blanc (Chester and Johnson, 1971a). Analogous to the samples of this study, the
$PM_{20}$ fraction of surface sediments of the piston cores RC05-57, RC05-60 and A180-44 also feature zeolites and
the surface sediments of core RCRC05-57 also traces of pyrophyllite (sepiolite belongs to the pyrophyllite group)
(Biscaye, 1964). Further, rutile was also present in the silt and sand fraction of Saharan dust sampled perpendicular
to the coast on the research vessel (Chester and Johnson, 1971a). Palygorskite was found in the clay fraction of
the surface sediment of sediment core GIK12329 (19° 22' N, 19°56' W) offshore Cape Blanc and is considered a
characteristic mineral of Saharan dust (Lange, 1975). The observed annual average C/K ratio (C/K=4) recorded
for the bulk size fraction of the trap samples was larger than the C/K ratio (C/K=0.3-1) recorded in the clay fraction
of surface sediment samples offshore Cape Blanc by Lange (1982). The disagreement may be due to the generally
larger percentage of kaolinite in the clay fraction compared to the silt fraction (Journet et al., 2014).

The dust samples of the site Iwik were further characterized by a dominance in quartz and feldspar (Fig. 8a). A
dominance in quartz has also been described for continental dust samples and soil samples collected in Mauritania
by Schütz and Sebert (1987). More than 20 papers published XRD data of northern African dust reporting quartz
as the main mineral in most dust samples (Scheuvens et al., 2013). Moreover, the continental sampling site is
surrounded by sand dunes which are rich in quartz minerals (Schlüter, 2008;Lancaster, 2013). A high quartz
content may therefore point to predominantly locally derived dust. The observed increase in micas and decrease
in quartz and feldspar observed for the marine samples relative to the Iwik samples (Fig. 8a) can be explained via
the preferential gravitational settling of the larger dust minerals quartz and feldspar during transport (Delany et al.,
1967;Chester and Johnson, 1971a;Glaccum and Prospero, 1980;Schütz and Sebert, 1987). A strong downwind
decrease in quartz content in Saharan dust was also observed by Korte et al. (2016).

**4.2 Mineralogy as a provenancing tool**
**4.2.1 Dust collected on land**
The back trajectories indicate that the dust sources for the dust collected in Iwik during winter were located NE
and E of the sampling site (Fig. 9a, Fig.10a,b), while those during summer were located W (within the PNBA) and
NNE of the sampling site (Fig. 11a, Fig. 12a,b). This is in accordance with a change in the dominant local surface
wind direction from NE in winter to NNE in summer (Fig. 2) and is also reflected in the clay-mineralogical
composition of the samples.
Generally, there is not much variability in the clay-mineralogical composition of the Iwik samples. The back
trajectories for the winter sample of 2014 indicate that the material was blown from the southwestern Reguibat
Shield (**PSA 2**) (Fig. 9a). The lack of palygorskite in this sample does not fit to the proposed bulk palygorskite
content (1-30 %) of **PSA 2** (Scheuvens et al., 2013). We argue that the sampled dust was most likely derived from



a single localized source instead of externally mixed sources of **PSA 2** during transport. The sample included the
characteristic minerals rutile and serpentine (Fig. 8b) which are usually a result of metamorphic processes (Deer
et al., 1992). Indeed, the western Reguibat Shield is composed of metamorphic and granitic rocks (Schofield et al.
(2006) and references therein) and the rocks are intruded by serpentinites (Schlüter, 2008). The sample was further
characterized by the highest quartz percentage among all samples (~ 50 %) (Fig. 8b). The sand dunes of the Azefal
sand sea which cover part of the southwestern Reguibat Shield might have sourced these quartz grains (Fig. 9a).
The sand dunes may have been fed by outcropping carbonate deposits at the northern rim of the Taoudeni Basin
via the NE-trade winds leading to anomalously high percentages of calcite in the sand dunes (Fig. 9). Thus, the
sand dunes may have also sourced the calcite present in the sample (Fig. 8b).
The winter sample of 2014-2015 was suggested to be sourced from sediments of the northern Senegal-Mauritania
Basin (**PSA 2**) (Fig. 10a) and the eastern rim of the Taoudeni Basin (**PSA 3**) (Fig 10b). The palygorskite content
of the sample (8 %) fits to the proposed bulk palygorskite content of **PSA 2** (Scheuvens et al., 2013). This points
to several externally mixed sources during transport instead of single local source. The sample was further
characterized by calcite and chlorite (Fig. 8b). The sediments in the northern Senegal-Mauritania Basin (Fig. 10a)
comprise Quaternary chalky horizons (Wissmann, 1982) which may have sourced the calcite. More likely, calcite
may have been derived from the Mesozoic carbonate sequences cropping out in the eastern rim of the Taoudeni
Basin (Bertrand-Sarfati et al., 1991) (Fig. 10b). A source area lying at the Algerian/Mali border was also suggested
for a chlorite and calcite bearing dust sample collected on the Canary Islands (Alastuey et al., 2005). The winter
dust sample trapped at site Iwik was further characterized by the lowest feldspar percentage (~ 5 %), highest mica
percentage (~ 40 %) (Fig. 8b) and lowest modal grain size (~ 38 μm) among all Iwik dust samples analysed for
mineralogy. The Stokes terminal settling velocity is smaller for platy particles than for spherical particles of similar
diameter (Santamarina and Cho, 2004). Therefore, a long-distance transport of dust from the eastern Taoudeni
Basin to Iwik may have resulted in a depletion in spherical quartz particles (Fig. 5a,b,c) and an enrichment in platy
mica particles (Fig. 5b).
The summer sample of 2013 was proposed to be sourced from the near-by northern Tidra Island (**PSA 2**) (Fig.
11a). Again, the absence of the mineral palygorskite is noteworthy which points to a single localized dust source.
The sample was further characterized by the mineral gibbsite (Fig. 8c). The northern Tidra Island is famous for
the local occurrence of west Africa's northernmost mangroves (Proske et al., 2008) which grow in humid and
warm climates. Humid and warm conditions are also beneficial for the formation of gibbsite which forms through
tropical weathering (Deer et al., 1992). Therefore, we argue that the soils of Tidra Island supplied the gibbsite
found in the sample. A localized small gibbsite maximum was outlined for the surface sediments offshore Cape
Blanc (Biscaye, 1964) which further supports the view that gibbsite is supplied from a local source. The sample
was further characterized by anomalously large moderately spherical quartz grains (Fig. 5c) emphasizing a short
travel distance of the dust.
The summer sample of 2014 was most likely sourced by sediments of the Western Sahara (**PSA 2**) (Fig. 12a,b).
The palygorskite content of the sample (5 %) matches with the proposed bulk palygorskite content of **PSA 2**
(Scheuvens et al., 2013). Hence, dust was supplied from several local dust sources of **PSA 2** which were mixed
during transport. The sample was further characterized by calcite and dolomite (Fig. 8c). Sediments outcropping





in the Western Sahara are composed of Tertiary sediments (Wissmann, 1982) with limestone deposits (Bosse and
Gwosdz, 1996) that may explain the calcite found in the sample (Fig., 12a). Upper cretaceous outcrops in the
Aaiun-Tarfaya Basin near Laâyoune comprise dolomites (Bosse and Gwosdz, 1996) and could have sourced the
dolomite found in the sample (Fig. 12b). A further evidence for dolomite-bearing dust transport from the Aaiun-
Tarfaya Basin is a local dolomite maximum outlined for the surface sediments offshore the Western Sahara
(Johnson, 1979). A Saharan dust sample trapped in NE Spain also contained dolomite and calcite and was related
to a source area lying in the Western Sahara (Avila et al., 1997).

**4.2.1 Dust collected at the marine sites**
The seasonal contrast in the dust transport patterns (high-level Saharan Air Layer vs. low-level Trades) potentially
leaded to strongly deviating dust sources for the material deposited in the marine trap samples. During winter, the
back trajectories indicated that the potential dust source areas were located NE of the sampling site (Fig. 9b, Fig.
10c,d), while those during summer were located NE, E and SE of the sampling site (Fig. 11b, Fig.12a,c,d,e). This
large variability in wind patterns can clearly be recognized in the clay-mineralogical compositions of the samples
throughout the seasons.
Considering the much larger catchment area of the traps, several localized dust sources may have been sampled
with the traps. As a result, the composition of the analyzed samples fit well to the bulk composition of the chosen
**PSA**. The back trajectories indicate that the winter sample of 2014 originated from the shoreline of the Western
Sahara (**PSA 2**) (Fig. 9b). The observed C/K ratio (C/K=1) and the palygorskite content (11 %) are in agreement
with the bulk compositional C/K ratio (C/K=0-1) and palygorskite content of **PSA 2** (Scheuvens et al., 2013). The
sample was further characterized by the presence of chlorite, kaolinite, smectite, garnet, anhydrite and fluellite
(Fig. 8b). The characteristic occurrence of garnet together with the highest quartz content (33 %, Fig. 8b) among
all CBi samples confirms a short transport distance of the trapped dust. Chlorite may be sourced from a small
coastal area where chlorite-rich fluvisols are found (Journet et al., 2014) (Fig. 9b). The mineral fluellite which is
a weathering product of phosphate may have been derived from outcropping phosphate deposits near the Bucraa
phosphate mine (Moreno et al., 2006) (Fig. 9). The occurrence of kaolinite in the sample is remarkable as this
mineral was not observed in high amounts in the soils underlying the back trajectories. However, the sample was
further characterized by a lack of feldspar (Fig. 8b), which tends to be hydrothermally altered to kaolinite (Deer et
al., 1992). The same process may explain the presence of anhydrite in this sample, although this mineral could
also originate from evaporites along the coast. Another explanation for the presence of kaolinite and smectite may
be the transport of these minerals from southern latitudes via the poleward-flowing undercurrent to the trap site
CBi (Fig. 1). Kaolinite and smectite were found in the clay fraction of the surface sediments off Senegal (Nizou et
al., 2011) and may have been brought into the ocean by the Senegal River, and redistributed by ocean currents
(Biscaye, 1964). The season of high Senegal River sediment supply is between July to October/November (Gac
and Kane, 1986). Assuming a mean speed of ~10 cm/s of the undercurrent (Mittelstaedt, 1991), it may take about
two months for the particles to travel a distance of ~500 km to the trap site CBi. This time delay might explain the
observed occurrence of these minerals in the trap samples during winter, but not during summer.

The back trajectories of the winter sample of 2014 to 2015 lead to the Reguibat Shield (**PSA 2**) (Fig. 10c) and
coastal Western Sahara (**PSA 2**) (Fig. 10d). The observed C/K ratio (C/K=0) and palygorskite content (1 %) fall





within the ranges of these minerals in **PSA 2** (Scheuvens et al., 2013). The sample was further characterized by
the mineral zeolite (Fig. 8b). Zeolites are formed from volcanic glass and tuff and form well-developed crystals in
basalts (Deer et al., 1992). Therefore, the source area of the zeolites may have been outcropping volcanic rocks in
the northern Taoudeni Basin (Fig. 10c). These rocks belong to mafic dikes and sills which are commonly basalts
with dotted patches of glass (Verati et al., 2005). An additional indication for a distant dust source may be the
lowest quartz content (4 %) among all samples (Fig. 8b). The CBi trap sample was further characterized by the
minerals sepiolite and smectite (Fig. 8b). Sepiolite belongs to the pyrophyllites which is a mineral that also may
be considered indicative of tropical weathering (Moore and Reynolds, 1989). Similar to the winter dust sample
recovered during 2014, sepiolite and smectite may have been derived from humid weathering on the wet shoreline
of the Western Sahara (Fig. 10d) or from current transport of clay particles from the Senegal River mouth.
Palygorskite-sepiolite mafic clays were found in soil samples of the Western Sahara (Moreno et al., 2006) which
may supports a Western Saharan source.

Based on the back trajectories, the summer sample of 2013 was suggested to be sourced from the Mauritanides
(**PSA 2**) (Fig. 11c). This is confirmed by the palygorskite content of the sample (2 %) (Scheuvens et al., 2013).
Outstanding minerals in this sample are chlorite and rutile (Fig. 8c). Outcrops in the Mauritanides west of the
Taoudeni Basin feature strongly metamorphosed rocks (Villeneuve, 2005) and greenschist facies (Dallmeyer and
Lécorché, 2012) which may have been the source of the rutile and chlorite.

The reconstructed source area of the summer sample of 2014 was the Pharusian belt (**PSA 3**) (Fig. 12c) and the
extrusive volcanics of the northern Taoudeni Basin (**PSA 2**) (Fig. 12d). The lack of palygorskite in the sample
does corroborate with **PSA 4** ('not detected') (Scheuvens et al., 2013) suggesting that the provenance of the dust
sample may be mainly confined to **PSA 4**. The sample was further characterized by zeolite and ferryglaucophane
(Fig. 8c). The dike swarms and sills of the northern Taoudeni Basin (Verati et al., 2005) (Fig. 10c, Fig. 12c) and/or
the basalts of the Fezzan uplift (Fig. 12e) may have sourced the zeolite. Indeed, zeolite was described as one of
the main secondary minerals in the basaltic rocks of the central Al-Harui Al-Abyas basalt flows (Abdel-Karim et
al., 2013) and in vesicles of the east Al Haruj basalts (Cvetković et al., 2010) of the Fezzan uplift. Traces of zeolite
were also detected in the Iwik sample during this sampling interval. It may be that the zeolite dropped out of the
high-altitude dust cloud and was subsequently transported via the surface trade winds to the continental trap site.
The presence of ferryglaucophane and the absence of feldspar and chorite in the sample indicates highly
metamorphous outcrops constituting the dust source. Therefore, the sample may have been additionally sourced
by the Pharusian belt (Fig. 12c) because blueschists were observed in Timétrine (Caby, 2014) and glaucophane
bearing eclogites in the Gourma fold and thrust belt north of Gao (Caby et al., 2008). The sample was further
characterized by the highest mica content (44 %) among all samples (Fig. 8c) supporting a large dust transport
distance.






**5. Summary and conclusions**
The fluxes, grain-size distributions and the mineral assemblages of the continental trap samples and oceanic
sediment trap samples were well comparable to the characteristics of Saharan dust reported for the region. The
following main findings were made:
- dust deposited on the continent was predominantly transported with the trade winds from proximal sources,
while dust deposited in the marine traps was transported with both the trade winds (winter, proximal) and in
the Saharan Air Layer (summer, distal) from proximal and distal sources
- the percentage of mica relative to the quartz content increased in the deposited dust with increasing transport
distance, most likely due to the platy shape of these minerals, which reduces settling
To conclude, the particle size and mineralogy of Saharan dust recorded in continental climate archives should be
interpreted differently with respect to paleo-environmental conditions compared to marine climate archives; the
on-land archive seems to reflect a much more local signal as compared to the regional signal that is recorded in
the marine sediments. Given the relationship between particle size and wind strength, we suggest that the particle
size in the continental archive in NW Africa may indicate the paleo-wind strength of the trade winds. This is an
intuitively logical conclusion, but it has not been demonstrated before so clearly. Finally, we have shown how the
mineralogical composition of the samples can be used for provenancing of dust particles found in both on-land
and marine dust archives.



















**6. Appendices**
**A1 Satellite RGB images**

In Fig. A1-4 satellite RGB true colour images are shown of the identified dust storms occurring during the sampling
interval of the samples analysed for dust provenance. On 31 July 2014 only few dust can be observed which
overlies the sampling location CBi (Fig. A2). This fits to the observed minor percentage of the mineral
ferryglaucophane (7 %) in the sample which was suggested to be sourced on 31 July 2014 from PSA 3. Zeolite,
which was more abundant (22 %) in the dust sample, was therefore most likely derived from PSA 4 due to the
major dust storm event occurring on 7 August 2014 (Fig. A3).

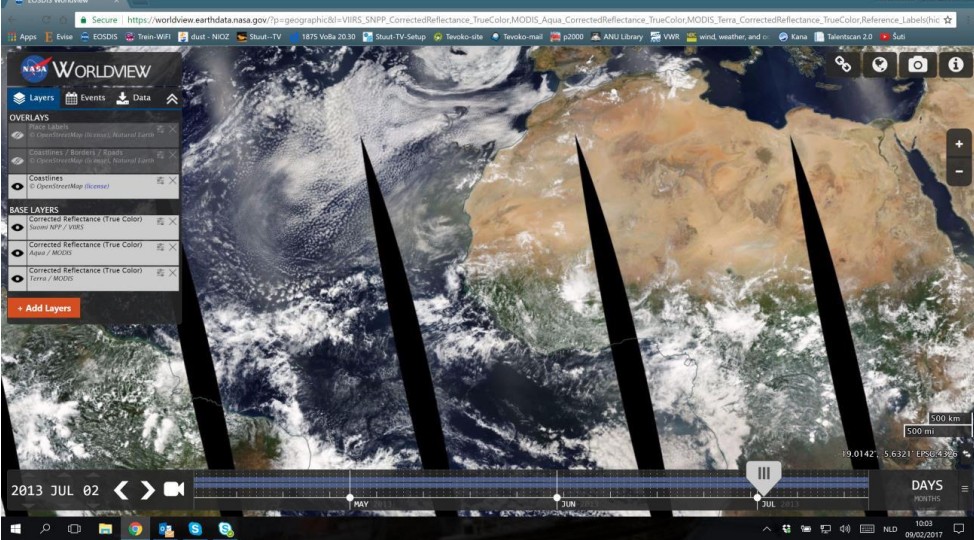


**Figure A1: Dust storm on 02 July 2013.**





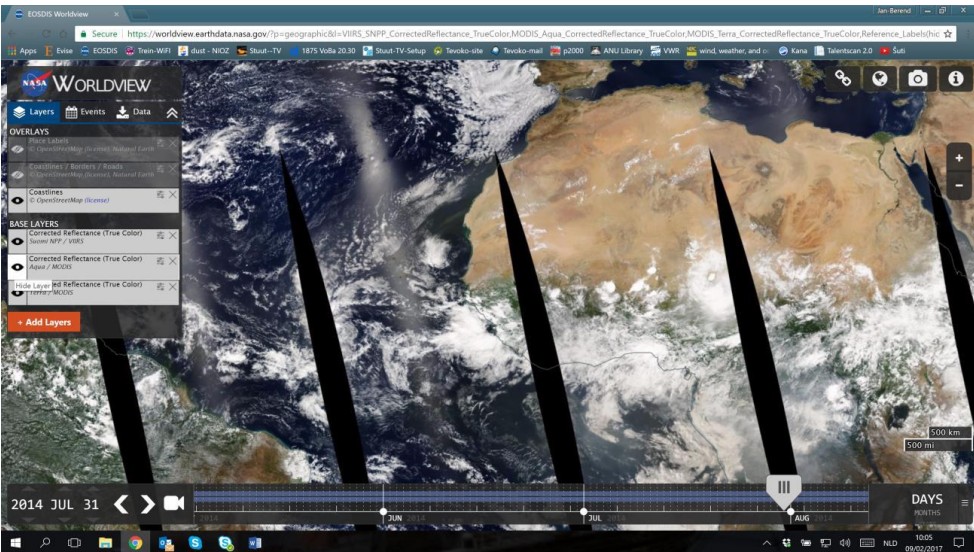


Figure A2: Dust storm on 31 July 2014.

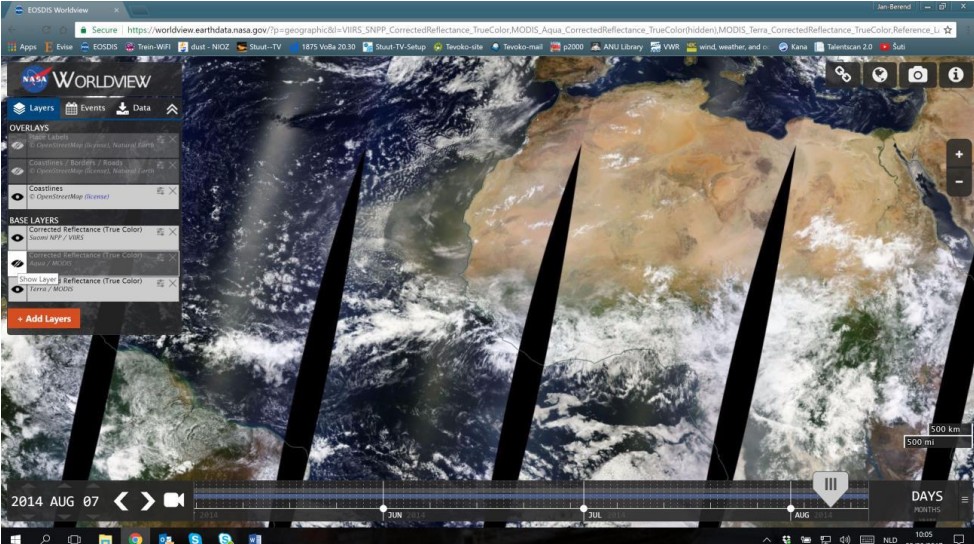


Figure A3: Dust storm on 07 August 2014.







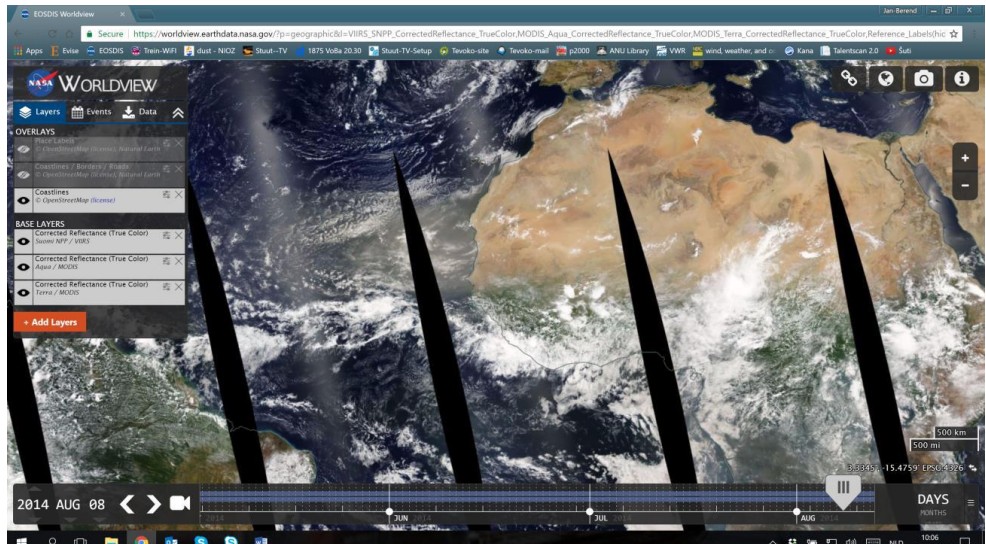

880                       **Figure A4: Dust storm on 08 August 2014.**


**A2 Four day back-trajectories**

In Fig. A5-8 the four day back-trajectories are shown calculated at the heights 3000 m, 4500 m and 5500 m ending
at site CBi. These high altitude back-trajectories were calculated for the identified summer days with dust storm
events (shown in Fig A1-4). On the one hand, a height of 4500 m was chosen by Skonieczny et al. (2013) in a dust
provenance study to represent the Saharan air layer (SAL). On the other hand, a height of 5500 m was chosen by
Ratmeyer et al. (1999a) in a dust transport study to represent the SAL. Maximum wind velocities within the SAL
are observed at a height of ~ 3 - 4 km in the area of the Cape Verde Islands during summer according to Carlson
and Prospero (1972). Therefore, we also plotted the back-trajectories at a height of 3000 m. In order to investigate
which air layer should be chosen for provenance studies, the back trajectories of the different heights were
compared.
The back-trajectories deviated slightly from each other regarding their direction and length. The back-trajectories
at 3000 m showed the most deviation. Further, the back-trajectories at 4500 m showed the best agreement with the
source areas and the minerals in the samples. Therefore, we chose to use the trajectories at 4500 m for provenance
studies according to Skonieczny et al. (2013).




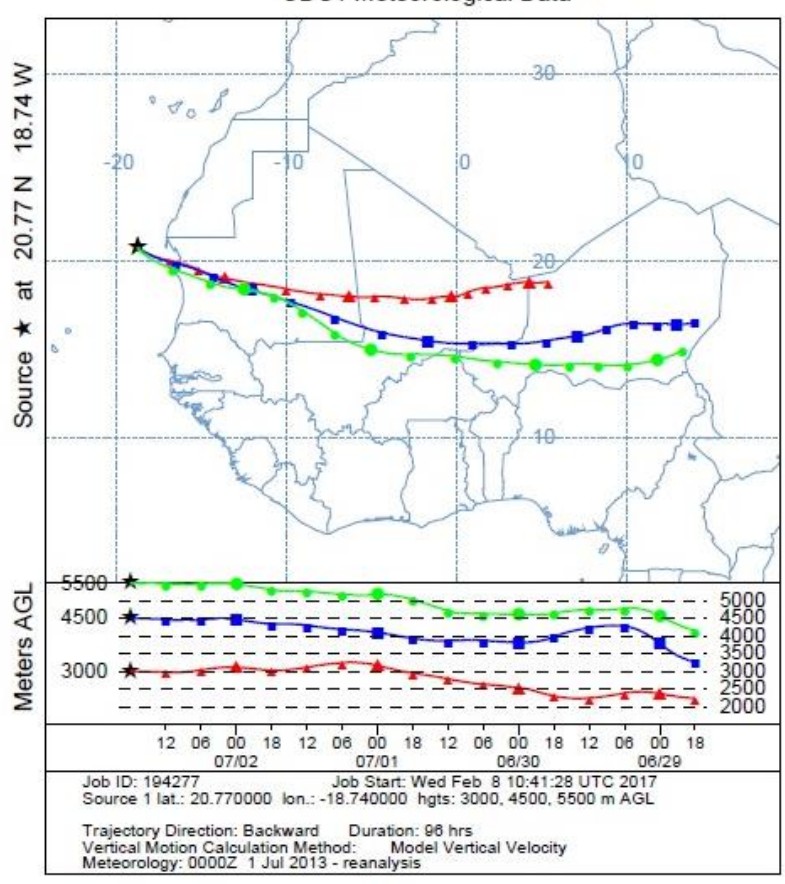


899          **Figure A5: Four day back-trajectories at a height of 3000 m, 4500 m and 5500 m on 02 July 2013.**






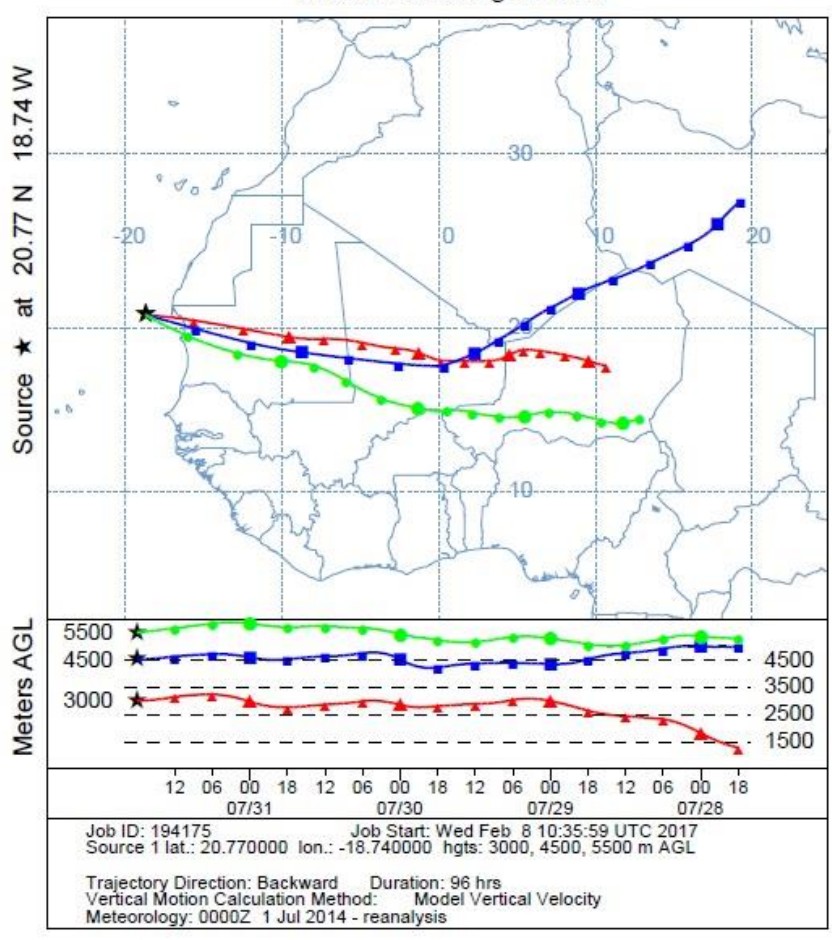


**Figure A6: Four day back-trajectories at a height of 3000 m, 4500 m and 5500 m on 31 July 2014.**

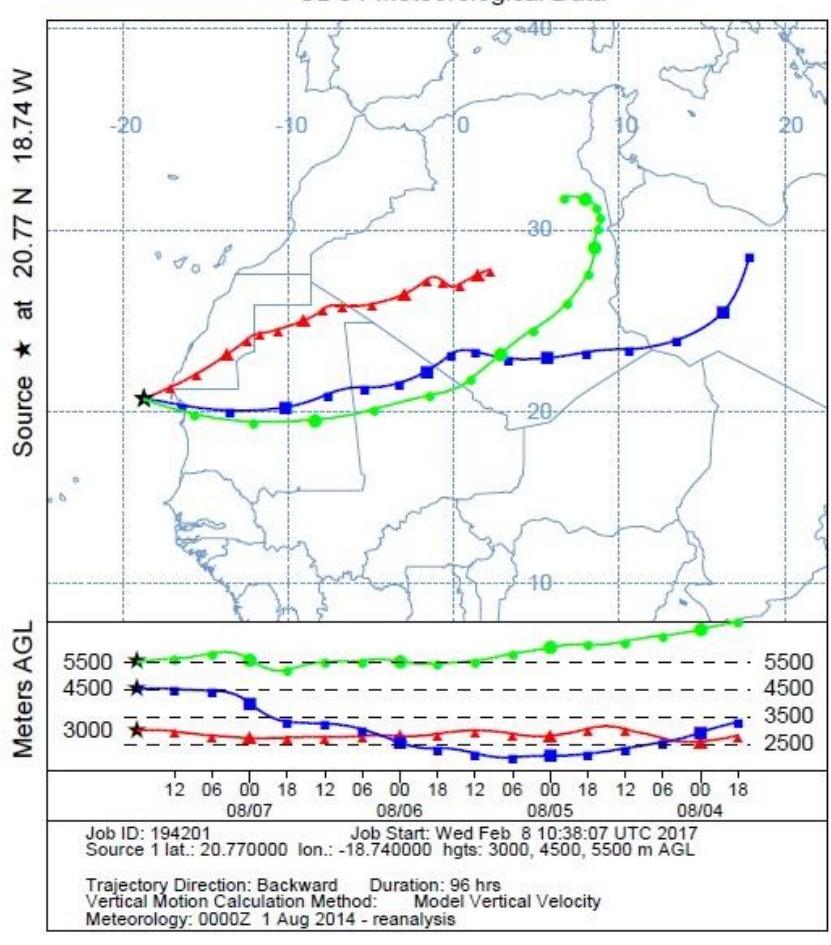


**Figure A7: Four day back-trajectories at a height of 3000 m, 4500 m and 5500 m on 07 August 2014.**



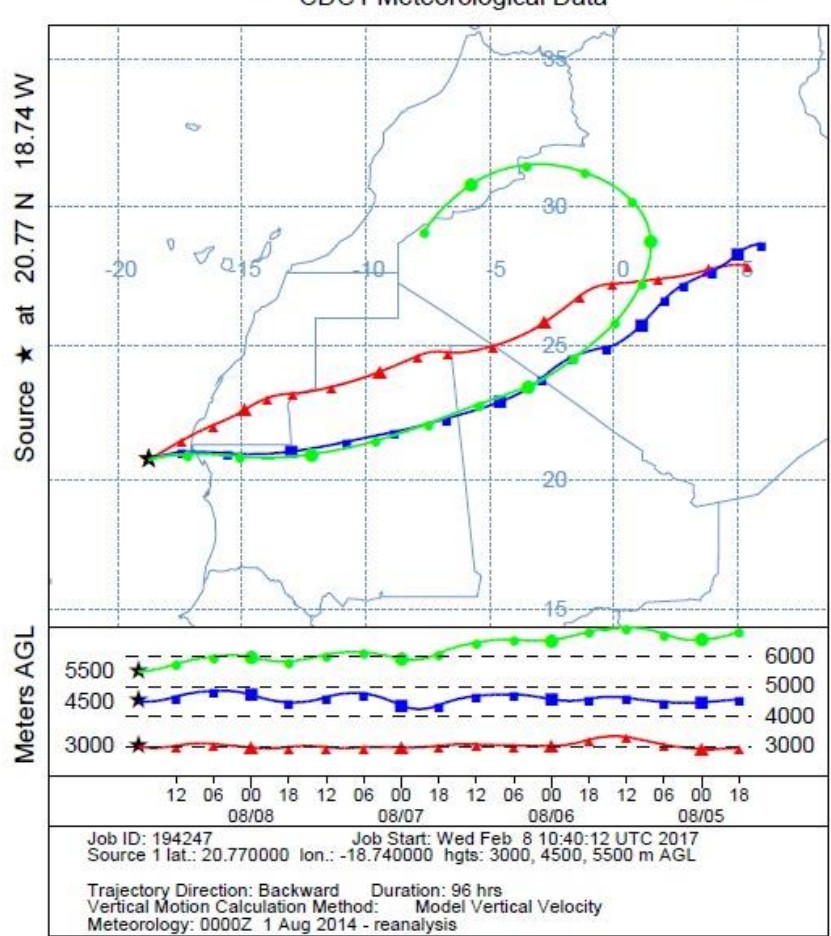


**Figure A8: Four day back-trajectories at a height of 3000 m, 4500 m and 5500 m on 08 August 2014.**

**7.  Supplement link**
The data can be accessed on www.pangaea.de.
**8.  Author contribution**
C. Friese carried out the particle size analysis of the sediment trap samples. H. van Hateren carried out the flux
and particle size analysis of the Iwik dust samples. G. Fischer provided the sediment trap samples and supervised
the flux analysis of the sediment trap samples. C. Friese prepared the samples for XRD analysis. C. Vogt carried
out the XRD analysis and was involved in the discussion of the results.  J.-B. Stuut managed the projects through





which dust-collecting buoy 'Carmen' was constructed and deployed, supervised the particle-size analysis and the
writing of the manuscript. C. Friese prepared the manuscript with contributions from all co-authors.
**9. Competing interests**
The authors declare that they have no conflict of interest.
**10. Acknowledgements**
We thank the captains, crews and scientific teams of the research cruises with RV Poseidon in 2013 (POS445),
RV Poseidon in 2014 (POS464) and RV Poseidon in 2015 (POS481), during which the sediment traps were
deployed and received. Further, we thank Marco Klann for preparing and splitting the sediment trap samples. Jan-
Berend Stuut acknowledges funding from ERC Grant 311152 DUSTTRAFFIC. Funding is acknowledged from
the German Science Foundation (DFG) through the DFG-Research Center/Cluster of Excellence 'The Ocean in
the Earth System'. We further thank Prof. Dr. Dierk Hebbeln and Dr. Ute Merkel for helpful and productive
scientific discussions.

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
