# Peer review of "Seasonal provenance changes of present-day Saharan dust"

_Atmospheric Chemistry and Physics, 2017_

## Referee Comment (RC1) · Anonymous Referee #1 · 31 Mar 2017

Review of the manuscript "Seasonal provenance changes of present-day Saharan dust collected on- and offshore Mauritania" by C. Friese et al., submitted to ACPD:

The work is an extensive analysis of different dust samples, sampled both on land and in the ocean. Samples were analyzed concerning different properties as grain size distributions and mineral composition, aiming at assigning source regions to the samples. Samples on land seem to be more influenced by local sources than oceanic samples, and it is indicated that long range transport decreases the average grain size of the deposited dust. This is all interesting and important information. However, being not a mineralogist, some passages were difficult to follow, and the overall structuring should be improved as well (see below). Also, occasionally statements seem to be

too far-fetched. So in total, my judgement is that the manuscript can become fit for publication in ACP, but only after major revisions.

General comments:

Several times in the manuscript, the impression is given that dust settling from the SAL starts to appear at the time when the air mass crosses the shore line, i.e., that SAL is a dust source above the Atlantic, only. This might just be formulated misunderstandingly, but in any case, dust will always settle from the SAL – it is just that on land, where nearby dust sources are present, they might be overwhelming. You have to go through the text and change the respective passages accordingly.

Your tables 1 and 2 are helpful, already, but I would have wished for an additional overview, showing which of the different locations contributed data for which kind of analysis. Also, all sites should be introduced together. E.g., Buoy Carmen is treated differently from CB and CBi, appearing for the first time in line 161, but not being shown in Fig. 1 and Fig. 2. Fig. 4 then shows the location of Iwik, CBi and Carmen and also of two new locations, but not the location of CB. This is all confusing, and a table could help to clarify that e.g., one place was only used for its information on meteorology, another one only for size distribution, etc. . Similarly, it is not clear to me what is gained by showing wind directions in Nouadhibou if information from that location is then dismissed due to local effects, unless you want to make it clearer that ground based sampling on land will not reflect long range transported dust (due to local sources and local winds) – but in this case, this needs to be discussed somewhat stronger. Consequently: if there are these local effects on land, will Iwik not be influenced similarly? This should be discussed, too.

For your determination of possible dust sources, you use trajectories in heights of 10m and 4500m. You chose 10m based on your even lower sampling heights. But still, trajectories in heights as low as 100m or below are always very prone to errors, and 4500m is so height that it might be above the heights of the SAL. Please at least

mention that and explain why you chose to calculate the trajectories at these heights, nevertheless. Alternatively, choose additional different heights (e.g., 100m and 2000m or so) – you could check, if they are similar to those you used. Adding them could add credibility to your work.

When you discuss the dust sources, the basic assumption seems to be that there is always one source with one mixture of minerals. But couldn't dust be emitted from areas with differing mineral composition in one dust storm (in fact, even the regions you selected as "major potential source regions" are not homogenous concerning the mineral compositions). My understanding of mineralogy might be too limited, here, but for me the attribution to the characteristics of the sampled dusts to the source regions seemed to include a lot of guessing. Again, tables could help, showing minerals that are present in the different "major potential source regions", and at the same time showing the minerals found in the different samples. This would certainly have helped me to follow your conclusions. Also, statistically, the number of dust sample cases that could be evaluated and the number of particles analyzed all are rather low. It is, no doubt, a lot of effort to do all this, but the resulting data should not be overrated. In this context, certainly the last sentence in the "summary and conclusion" (line 843) (and some other sentences as well) is formulated too strongly.

Specific comments:

Line 67-69: Is this statement taken from one of the papers cited in the text close by? Which one? And in general: How far offshore can regional / local dust emissions be found? (Or, in other words, how large is the "footprint" of a dust source when it does not emit the dust to large heights?)

Line 82: Do you mean your Fig. 1 or the Fig. 1 from Scheuvens et al.? Clarify.

Line 383: Arkeiss and Iwik are not oceanic, are they? Rephrase.

Line 432 - 436: This discussion about dust fluxes is confusing, and I suggest to completely remove it, also because (see next comment):

Line 438: Not surprisingly, the sampling heights has influences the results. This prohibits a direct comparison of mass concentrations and such parameters between the different sampling types (on mast and in traps in the water), while comparing shapes of size distributions and dust compositions might be possible. This should be discussed in the text somewhere. It is mentioned at line 607 again, but also not discussed. Based on this, a comparison of fluxes between the stations makes no sense.

Line 450: The bi-modal distribution of Iwik 13-14 is shown in Fig. 6c, right? Add this information to the text.

Fig. 6: There is a change in sequence between the two captions, which I found disturbing. I'd prefer to always list - CBi - CB - Iwik. Also: maybe put the black data on top of red one, in the plot, otherwise they are hard to see, or make a mix, so that at least some of the black ones are visible a bit better.

Line 452: Explain what you mean by "the sorting".

Figure 7 a) and b): Why was a different wind speed chosen in a) (> 5 m/s) and b) (> 6.5 m/s)? Also: there is a frame around Fig. 7 (on 3 of the 4 sides) that has to be removed for the final version.

Line 527: Why are there 2 back trajectories, only? Were they only made for the time of the dust event? Then why 2 and not only 1? This has to be clearly explained (this also is valid for Fig. 10, 11, and 12, where different numbers or trajectories are shown.)

Line 538: Just out of curiosity: does sea spray (i.e., the Cl in it) engrave the analysis of chloride?

Figure 9: The line for the 4500m back-trajectory for CBi is in the caption, but not in the plot. But it could be interesting to see it in here, as the dust transported from further away at higher altitudes might also contribute. This comment is valid also for Fig. 10.

Line 551: Replace "due to the" by ""as there are".

Line 596 ff: Was this for PM10 or total particulate mass? What exactly do you mean by "annual average dust concentrations"? The average per dust storm, or really the all-time average of dust, or something else? Please explain. Also: It is not clear to me how the monitoring of less dust events in Morocco due to the shorter sampling time can affect an average value, unless this average is "per year".

Line 606-607: If the difference is due to different sampling techniques, does it make sense to do such a comparison as you present it here?

Line 608: In which time span does 1% drop out of a moving dust cloud? Per day, per year, ever (certainly not), . . .?

Line 613-614: Replace "in the following" by "as follows".

Line 638: You could stress stronger that a bi-modal size distribution (as observed e.g., at Iwik in Fig. 6c) be indicative for nearby sources. Different size modes typically indicate different sources!

Line 641: Not only the wind speed, but also the heights into which dust was emitted, will influence how far it can be transported.

Line 648: I have a hard time imagining how precipitation droplets (with typically high fall velocities) enter your sampler in noticeable amounts. Or do you suggest here that precipitation is formed, falls, evaporates during its fall and leaves these dust particles behind at lower altitudes which can then be sampled? If this is so, mention it.

Line 692: Prior to this line, you explain that quartz seems to be present "everywhere" in North Africa. If this is the case, a high quartz content cannot be used to assume that it is mainly locally derived dust.

Line 736: This argument can only be correct if only this one nearby source does not have the mineral palygorskite in it, while it would be present in ALL other sources. Also:

not finding palygorskite might simply be an issue of low sampling statistics – can you exclude this? So overall: Can this text in the manuscript really be stated like this? This is also connected to what you state in line 709, so check this location for consistency, too.

Line 784 ff: How fast do dust particles sink in the water – can they travel 500km and still be collected at the sampling site CBi? Or, in other words, how large is the catchment area of a trap? You mention 40km * 40 km above in the text yourself for the upper trap? Therefore, it seems that the here described time delay cannot be used as an explanation.

Line 833-835: For continental sampling, rather than using the word "trade wind", the word "nearby sources" would be more appropriate. Also, the discrimination between dust particles transported in the SAL or by the trade winds is awkward. The trade winds transport the SAL, while the SAL is the source of particles that deposit from air masses that are moved by the trade winds. In this sense, "trade wind" and "SAL" both contribute together and should not be separated. The formulation needs rewording. Also check the whole text to remove respective inconsistencies.

Line 836-837: This result was only presented in bypassing (line 731 – or am I missing something), and I was quite surprised to see this as one of the main findings. On the other hand, changing grain sizes with distance to sources are not mentioned at all. I suggest to really reconsider which of your results are worth mentioning here, as for some readers, abstract and summary might be all they will ever look at.

Line 842 ff: The wind at the sampling location might differ from the wind in the source region, if the latter is far away, and the sizes of dust particles present in the source region will influence the grain sizes that can be deposited as well. There might still be a connection between wind speed and transported grain size, but it should be discussed in a broader sense, including (or at least mentioning) the points I raise in the previous sentence.

---

## Referee Comment (RC2) · Anonymous Referee #2 · 31 Mar 2017

In the manuscript, "Seasonal provenance changes of present-day Saharan dust collected on- and offshore Mauritania" the authors present data from sediment traps at multiple depths off the coast of Mauritania and surface collection sites close to the coast. They attempt to determine location and seasonality in potential source regions for different case studies based on the mineralogy of the samples and back trajectory analysis.

The description of the methodology and measurements is comprehensive and well thought out. I cannot speak to the specifics of the measurement methodology but have added minor comments and clarification requests below. Unfortunately, the broader context and the scientific developments are lacking in the paper. The measurements

are clearly valuable and should be published. The analysis of collected samples and the potential source regions is thorough in a qualitative sense. However, the useful scientific conclusions are not clear. This is indicated by the abstract that reads more like and introduction, the long, subjective discussion, and the relatively sparse summary and conclusions. For example, the last paragraph of the manuscript states that sediment records from land and ocean are likely to sample different source regions, based on the measurements showing more local sources over land. This could be an interesting point, but without further analysis (firmer understanding of the sources, dependence on the particular measurement site) the conclusion that sources are more likely to be local on land than at an ocean site further downwind seems common sense. I'm also not convinced that the atmospheric and sediment trap data should be presented side by side based on the difference in collection methodology and catchment. I think the authors need to consider how to better frame the important measurements presented in this manuscript, by presenting the data in a way that is easier to compare with other dust deposition and concentration measurements and a more thorough back trajectory analysis that answers the questions laid out in the introduction.

The choice of questions (lines 76-79) is a little strange. For example, (1) why would one not expect there to be seasonality in the deposition when we know that there is seasonality in winds leading to dust emission and transport? (2) This is an interesting question, but how dependent is this on the specific locations chosen? (3) This is very similar to question (1). (4) This is a good question but is only tackled in a qualitative way in the manuscript. I think laying out the questions to be answered is a good format; however, they currently seem like an afterthought and they should be returned to explicitly in the summary/conclusions.

While the back trajectory analysis is interesting, it is rudimentary. The choices of back trajectory heights appear to be arbitrary, I did not find a reason for the choice of 4500 m and 10 m is understandable, but those trajectories will be highly uncertain. A larger ensemble of back trajectories from different altitudes and start points would better quantify

the likelihood of dust sources and also help represent the uncertainties in back trajectories that pass so close to the surface. Further, including surface wind reanalyses (and other meteorology) in a comprehensive analysis of the mineralogy measurements and back trajectories, a more rigorous statistical analysis of likely source regions could be undertaken. A statistical approach would also provide a framework to analyze future measurements, rather than the case-by-case methodology shown. This would also help reduce the speculatory nature of the discussion of sources in Section 4.2.

Are the distributions really unimodal in Figure 6, as stated on line 620? I can see the finer mode and sometimes a third mode in there. Coming from an atmospheric modeling perspective, the mode between 1-10um is of great interest and it is a shame that this is not discussed more. From an atmospheric perspective, the value of this work could be increased by presenting more information on the finer dust particles. Models are always in need of aerosol size distribution measurements for evaluation of dust emission, transport and deposition. It would also be useful to have the size distributions presented as dM/dlnD or dV/dlnD to allow for comparison with model simulations and other measurements.

To summarize, I think:

- The measurements and presentation of the results are of great value

- Determining dust sources could benefit greatly from an analysis of surface winds from meteorological reanalyses to accompany the back trajectories.

- The summary and conclusions seems like an afterthought in the current presentation. Consider expanding this to crystallize the findings of the research better.

- The questions to be answered that are set out in the introduction are vague. It is useful to set out the motivational questions for the manuscript in this way, but I think more time should be put into the questions to be addressed and then ensuring that you return to these points in the discussion/summary/conclusions.

Minor Comments

line 14-15, 34 "Environmental parameters" is non-descriptive. Please revise. Line 34 could be deleted.

line 95 "In the following," - add comma

line 131 - Haboobs are normally defined as the dust storm from evaporatively driven cold pool outflow from convective events, not low-level jets

line 161-163 - this paragraph seems disconnected from the rest of the section

line 222 and 235 repeat

line 247 - why is 2xCorg removed from the total mass? Is this a general scaling from organic carbon to organic mass?

line 350 - I think the long url links should go in the data availability section rather than in the text. Write out the usage but simply reference the data section rather than talking about downloading files in the manuscript.

line 357-358 - this repeats the previous sentence, condense.

line 377 - "ground station" could be misinterpreted as on land, consider "surface station"

line 461 - define "well sorted"

Figure 6 - These are not really unimodal, as referred in the text (line 620)

line 602 - which year?

line 608 - is horizontal flux a useful metric to compare with ocean deposition?

A minor issue, but there are formatting errors with brackets on the references throughout that need fixing.

---

## Author Response (AR1)

Carmen Friese

Leobener Str.

28359 Bremen

Germany

Tel.: ++49-(0)-421-218-65584

Email: cfriese@marum.de

Dr. Schwarz

NOAA ESRL Chemical Sciences Division

Broadway, R/CSD6

Boulder, CO 80305 USA

**Submission of revised manuscript**

Dear Dr. Schwarz,                                                                                      16 June 2017

thank you for the time you spent on assessing the manuscript *'Seasonal provenance changes of present-day Saharan dust collected on- and offshore Mauritania'* by C. Friese et al.. We received two very good (anonymous) reviews with detailed constructive comments which helped to improve the interpretation as well as structure and clarity of the manuscript. Therefore, we would also like to thank the reviewers for the time they spent on evaluating the manuscript and for elaborating their comments.

We received a number of similar comments from both reviewers. One dealt with the number of trajectories plotted in order to derive the likely source areas of the individual dust samples. Both reviewers recommended to show additional back trajectory heights because the shown back trajectory at 10 m may be prone to errors and the back trajectory at 4500 m may be higher than the SAL. We chose these back trajectory heights based on the studies of Skonieczny et al. (2013) and Stuut et al. (2005) who used these heights to represent the SAL and the trade-winds respectively. To improve the identification of likely dust sources, we additionally plotted back trajectories at a height of 100 m and 3000 m. This especially improved the determination of the sources of dust transported at low-level.

The reviewers asked for an explanation of 'the sorting' of a grain- size distribution and a definition of 'well-sorted'. The sorting is a word used by sedimentologists to characterize the grain-size distribution: well-sorted refers to similar sizes and thus a low standard deviation of the grain-size distribution. To improve the clarity of the manuscript, we added this explanation to the text.

Moreover, both reviewers suggested to exclude the comparison between the atmospheric dust fluxes at the continental site and the dust deposition fluxes at the oceanic sites. This of course makes sense because atmospheric dust fluxes are higher by 2 – orders of magnitude compared to depositional fluxes and therefore it is difficult to execute a quantitative comparison. Following the reviewers' suggestion, we removed the comparison which further improved the content of the manuscript.

Please find each of the reviewers' individual comments and our revisions based on the reviewers' comments in the attached supplements as well as in the revised manuscript. The reviewer's comments are given with black normal text and our reply to the comments are given with black italic text. We used different colours for the revisions in the manuscript in response to the respective reviewer (referee #1 = red, referee #2 = blue).

Thanks again for your time and we look forward to hearing from you.

On behalf of all co-authors,
yours sincerely,

Dr. Carmen Friese

**References**

Skonieczny, C., Bory, A., Bout-Roumazeilles, V., Abouchami, W., Galer, S. J. G., Crosta, X., Diallo, A., and Ndiaye, T.: A three-year time series of mineral dust deposits on the West African margin: Sedimentological and geochemical signatures and implications for interpretation of marine paleo-dust records, Earth and Planetary Science Letters, 364, 145-156, http://dx.doi.org/10.1016/j.epsl.2012.12.039, 2013.

Stuut, J.-B. W., Zabel, M., Ratmeyer, V., Helmke, P., Schefuß, E., Lavik, G., and Schneider, R. R.: Provenance of present-day eolian dust collected off NW Africa, Journal of Geophysical Research, 110, 10.1029/2004JD005161, 2005.

**Comment of Referee #1:**

The work is an extensive analysis of different dust samples, sampled both on land and in the ocean. Samples were analyzed concerning different properties as grain size distributions and mineral composition, aiming at assigning source regions to the samples. Samples on land seem to be more influenced by local sources than oceanic samples, and it is indicated that long range transport decreases the average grain size of the deposited dust. This is all interesting and important information. However, being not a mineralogist, some passages were difficult to follow, and the overall structuring should be improved as well (see below). Also, occasionally statements seem to be too far-fetched. So in total, my judgement is that the manuscript can become fit for publication in ACP, but only after major revisions.

**General comments:**

**Comment of Referee #1:**

Several times in the manuscript, the impression is given that dust settling from the SAL starts to appear at the time when the air mass crosses the shore line, i.e., that SAL is a dust source above the Atlantic, only. This might just be formulated misunderstandingly, but in any case, dust will always settle from the SAL – it is just that on land, where nearby dust sources are present, they might be overwhelming. You have to go through the text and change the respective passages accordingly.

**Reply to comment:**

*This is a good point. Dust already settles on land from the SAL and we formulated this unclearly. We agree, that the passages in which this happened need to be revised. Therefore, we changed all the sentences in which such a misunderstanding could lead to a false view on the dust transport.*

**Changes made in manuscript:**

We modified the following sentences (page and line numbers refer to the revised manuscript):

Page 1, lines 19-20:

'In continental dust source areas dust is also transported in the SAL, however the predominant dust input occurs from nearby dust sources with the low-level trade winds.'

Page 5, lines 150-153:

'Low-level N trade winds blow and transport dust in coastal Mauritania year-round (National Geospatial-Intelligence Agency, 2006). Saharan dust is transported on- and offshore within the 'Saharan air layer' (SAL) at an altitude of about 3 km (Prospero and Carlson, 1970;Carlson and Prospero, 1972;Prospero and Carlson, 1972;Diaz et al., 1976).'

**Comment of Referee #1:**

Your tables 1 and 2 are helpful, already, but I would have wished for an additional overview, showing which of the different locations contributed data for which kind of analysis. Also, all sites should be introduced together. E.g., Buoy Carmen is treated differently from CB and CBi, appearing for the first time in line 161, but not being shown in Fig. 1 and Fig. 2. Fig. 4 then shows the location of Iwik, CBi and Carmen and also of two new locations, but not the location of CB. This is all confusing, and a table could help to clarify that e.g., one place was only used for its information on meteorology, another one only for size distribution, etc. .

**Reply to comment:**

*We agree that the structure of the manuscript would greatly improve when showing all sites together and when inserting a table showing which analysis has been done at each site. Therefore, we modified figure 1 by inserting all study sites together. Further, we inserted a new table 1 which gives an overview of the study sites with information on the respective location, samples, analyses and data.*

**Changes made in manuscript:**

We inserted the following sentences:

Page 3, line 73:

'and with the scientific dust-collecting buoy 'Carmen''

Page 3, lines 82-88:

'In Fig. 1 the location of the study sites and the North African dust sources are displayed. The dust-collecting buoy 'Carmen' (~21°15' N, ~20°56' W) and the sediment trap mooring site CB (~21°16' N, ~20°48' W) are virtually at the same position ~200 nautical miles offshore Cape Blanc. Sediment-trap station CBi (~20°45' N, ~18°42' W) is located ~ 80 nautical miles offshore Cape Blanc. The continental dust collector Iwik (~19°53' N, ~16° 18' W) and the meteorological station Arkeiss (~20° 7' N, ~16° 15' W) are located in a major potential dust source area (**PSA 2**) in the Parc National de Banc d'Arguin (PNBA) near Iwik and near Arkeiss in Mauritania. A further meteorological station is positioned in the **PSA 2** in Nouadhibou (~20° 55' N, ~17° 1' W) in the Western Sahara.'

We shifted the following sentence to:

Page 5, lines 118-119:

'The local soils surrounding the dust collector site Iwik are composed of sandy deposits often rich in fossil shells and partly cemented by lime (Einsele et al., 1974).'

We modified Fig. 1:

[Figure]

**Figure 1: Map of the study sites under investigation: the scientific buoy Carmen as well as the sediment trap moorings CB and CBi offshore Cape Blanc, the MWAC dust collector onshore near Iwik and the ground stations near Nouadhibou and Arkeiss (shapefile of the surface lithology and the geological provinces: downloaded from the USGS website http://rmgsc.cr.usgs.gov/ecosystems/africa.shtml#SL and http://certmapper.cr.usgs.gov/geoportal/catalog/main/home.page, major potential dust source areas: redrawn from Scheuvens et al. (2013), ocean currents: redrawn from Mittelstaedt (1991)).**

We inserted the following sentences and table:

Page 7, lines 190-199:

'In Table 1 an overview of the material and methods employed for each study site is presented. Bulk sediment samples were obtained at the sites CB and CBi and dust samples at the sites Carmen and Iwik. All samples were analyzed for particle size and dust flux with the exception of the site Carmen, of which only dust particle size was analyzed. Only the sites CBi and Iwik were analyzed for mineral assemblages and only the samples of the site Iwik were used for microscopic investigation. Meteorological sensors were available for the stations Carmen, Iwik and Arkeiss, while for the site Nouadhibou meteorological data was downloaded online. TRMM precipitation data was downloaded online for all sites except for the site Nouadhibou.

**Table 1: Overview of the material and methods employed at each study site. '**

| Study site | Lat./lon. | Samples | Analysis | Meteorological sensor and data | Downloaded meteorological data |
|---|---|---|---|---|---|
| **Carmen** | ~21°15' N, ~20°56' W | 1 MWAC sample | particle size | Vaisala WXT520: wind direction + speed, precipitation | TRMM 3B42: precipitation |
| **CB** | ~21°16' N, ~20°48' W | 38 sediment trap samples | lithogenic fluxes, particle size | - | TRMM 3B42: precipitation |
| **CBi** | ~20°45' N, ~18°42' W | 38 sediment trap samples | lithogenic fluxes, particle size, mineral assemblages | - | TRMM 3B42: precipitation, HYSPLIT back trajectories |
| **Iwik** | ~19°53' N, ~16° 18' W | 24 MWAC samples | microscopy, dust fluxes, particle size, mineral assemblages | Davis 6250 Vantage Vue: wind direction + speed | TRMM 3B42: precipitation, HYSPLIT back trajectories |
| **Nouadhibou** | 20° 55' N, 17° 1' W | - | - | - | Wind direction + speed |
| **Arkeiss** | ~20° 7' N, ~16° 15' W | - | - | Davis 6250 Vantage Vue: precipitation | TRMM 3B42: precipitation |

**Comment of Referee #1:**

Similarly, it is not clear to me what is gained by showing wind directions in Nouadhibou if information from that location is then dismissed due to local effects, unless you want to make it clearer that ground based sampling on land will not reflect long range transported dust (due to local sources and local winds) – but in this case, this needs to be discussed somewhat stronger. Consequently: if there are these local effects on land, will Iwik not be influenced similarly? This should be discussed, too.

**Reply to comment:**

*We agree that the results on the meteorological data at station Nouadhibou should be mentioned in the discussion. The site served as an example for how local effects may influence the observed wind patterns. At site Iwik the synoptic change in the predominant trade wind direction can be clearly observed, while at other sites like Nouadhibou local effects bias an observation of synoptic wind patterns. This should also be kept in mind when choosing a site for paleoenvironmental reconstructions using sediment core records.*

**Changes made in manuscript:**

We inserted the following sentences:

Page 35, lines 768-771:

'The variability of the mineralogical composition of dust sampled at site Iwik could be related to the synoptic scale change in the surface trade wind direction. However, meteorological data from nearby sites like e.g., Nouadhibou demonstrate that local effects like the topography exert a strong influence on observed wind directions at ground level. (Fig. 2).'

**Comment of Referee #1:**

For your determination of possible dust sources, you use trajectories in heights of 10m and 4500m. You chose 10m based on your even lower sampling heights. But still, trajectories in heights as low as 100m or below are always very prone to errors, and 4500m is so height that it might be above the heights of the SAL. Please at least mention that and explain why you chose to calculate the trajectories at these heights, nevertheless.
Alternatively, choose additional different heights (e.g., 100m and 2000m or so) – you could check, if they are similar to those you used. Adding them could add credibility to your work.

**Reply to comment:**

*We chose the altitudes at which back trajectories were calculated on the basis of previous studies to which we compare our data. For example, Skonienczny et al., 2013 use 4500m to study the SAL. Following the reviewer's suggestion, we now also included additional back trajectories at additional altitudes, which made the observed patterns much clearer. Back-trajectories of the heights 100 m and 3000 m were added to figures 9-12. This improved the detection of the likely dust source areas and the results and interpretation were adjusted.*

**Changes made in manuscript:**

We modified Fig. 9-12:

Page 24, line 563:

[Figure]

Page 26, line 585:

[Figure]

Page 27, line 598:

[Figure]

**Lithology**

| | | |
|---|---|---|
| Carbonate | Ultramafic | Alluvium - beach, strand, coastal dune |
| Karst | Extrusive Volcanic | Alluvium - Saline |
| Non-Carbonate | Colluvium | Alluvium - Gypsum |
| Metasedimentary | Hydric - Organic | Alluvium - Other |
| Alkaline Intrusive Volcanic | Aeolian Sediments | Volcanic - Ash, Tuff, Mudflow |
| Silicic | Alluvium - Fan Deposit | Water |
| Metaigneous | Alluvium - Fluvial | |

**Back trajectories**    ● Iwik    ● CBi

— 10 m   — 100 m   — 10 m   — 100 m   -- 3000 m   ▪-▪ 4500 m

**PSAs:** ▢ PSA 1-5

**Clay Mineralogy**

No data    Calcite >8.9 %    Kaolinite >29 %    Chlorite >4.1 %

**Mineralogy**

| | | | | |
|---|---|---|---|---|
| Kaolinite | Feldspar | Serpentine | Sepiolite | Gibbsite |
| Chlorite | Mica | Rutile | Anhydrite | Dolomite |
| Calcite | Amphibole | Zeolite | Smectite | Other minerals |
| Quartz | Palygorskite | Fluellite | Garnet | ★ Ferryglaucophane |

(c) 02.07.2013

[Figure]

We modified the following sentence to:

Page 14, lines 373-377:

'Four-day back trajectories at altitudes of 10 (following Stuut et al. (2005)), 100, 3000, 4500 (following Skonieczny et al. (2013)) and 5500m were calculated ending at the dust collector site Iwik (19°52' N, 16°17' W) and at the proximal marine trap site CBi (20°46',18°44' W) using the Hybrid Single Particle Langrangian

Integrated Trajectory (HYSPLIT) model (Stein et al., 2015) and the reanalysis dataset (2.5° spatial resolution) on the NOAA website at http://ready.arl.noaa.gov. '

We modified the following sentences:

Page 22, lines 546-548:

'Four heights, 10, 100, 3000 and 4500 m were chosen to cover both low- (trades) and high-level (SAL) dust transport.'

Page 22, lines 551-552:

'The back-trajectories at 5500 m can be found in the supplements. '

Page 23, lines 561-562:

'Thus, the source area of the samples was most likely the chlorite and kaolinite rich sediments located near the Bou Craa phosphate mine in the Western Sahara (Fig. 9b).'

Page 25, lines 581-584:

'A further source area might be the southern shoreline of the Western Sahara in which chlorite depleted sediments are situated (Fig. 10d). The presence of the mineral kaolinite in this marine winter sample may be explained by a kaolinite-rich source area lying in the southern Senegal-Mauritania Basin (Fig. 10e).'

**Comment of Referee #1:**

When you discuss the dust sources, the basic assumption seems to be that there is always one source with one mixture of minerals. But couldn't dust be emitted from areas with differing mineral composition in one dust storm (in fact, even the regions you selected as "major potential source regions" are not homogenous concerning the mineral compositions). My understanding of mineralogy might be too limited, here, but for me the attribution to the characteristics of the sampled dusts to the source regions seemed to include a lot of guessing.

**Reply to comment:**

*Of course, your argument is correct. Individual dust outbreaks may result from multiple dust source areas and the source areas are not homogenous with respect to the mineralogical composition. However, in order to be able to distinguish between the different PSA's we chose typical minerals that are characteristic for the different PSA's as defined by Scheuvens et al., 2013.*

**Comment of Referee #1:**

Again, tables could help, showing minerals that are present in the different "major potential source regions", and at the same time showing the minerals found in the different samples. This would certainly have helped me to follow your conclusions. Also, statistically, the number of dust sample cases that could be evaluated and the number of particles analyzed all are rather low. It is, no doubt, a lot of effort to do all this, but the resulting data should not be overrated. In this context, certainly the last sentence in the "summary and conclusion" (line 843) (and some other sentences as well) is formulated too strongly.

**Reply to comment:**

*Yes, indeed it is not easy to follow how the characteristic minerals of the individual samples were assigned to the PSA's and source rocks and deposits. Therefore, we included a table displaying information on the sampling interval and characteristic minerals of each sample. We further inserted information on the bulk mineralogical composition of the chosen PSA according to Scheuvens et al., 2013 and information on the characteristic source rocks and deposits of the chosen source areas. When looking at the table, it should be clearer now how the source areas were chosen based on the characteristic minerals of the samples.*
*We further agree that the last sentence in the summary and conclusion is formulated too strongly based on the fact that the number of samples is rather low. Therefore, we removed the sentence.*

**Changes made in manuscript:**

We inserted a table:

Page 34, lines 760-764:

'In Table 7 an overview of the chosen dust source areas for the site Iwik and CBi is given together with the characteristic minerals of the samples that may be used as a tracer for the source area. In the following subsections the identification of the source areas and mineralogical tracers is described in detail.

**Table 7: Overview of the chosen source areas and the tracer minerals of the individual samples together with the given characteristics of the source areas according to literature. '**

| Sampling interval | Characteristic minerals of sample | Chosen dust source area | Bulk mineralogical composition of chosen PSA [16] | Characteristic source rocks and deposits of chosen source area |
|---|---|---|---|---|
| **Iwik** | | | | |
| 15.02.-15.03.14 | *Rut, Serp, Cc | **PSA 2**: Reguibat Shield | C/K = 0.0–1.0 *Pal: 1-30 wt% | Metamorphic and granitic rocks [1] Serpentinites [2] |
| 15.12.14-18.01.15 | *Cc, Chl, Pal (8 wt. %) | **PSA 2**: Senegal-Mauritania Basin | C/K = 0.0–1.0 *Pal: 1-30 wt% | Chalky horizons [3] |
| | | **PSA 3:** Eastern Taoudeni Basin | C/K = 0.2–0.9 *Pal:1-5 wt% | Carbonate sequences [4] |
| 24.06.-15.07.13 | *Gib | **PSA 2:** Tidra Island | C/K = 0.0–1.0 *Pal: 1-30 wt% | Gibbsite maximum offshore Cape Blanc [5] |
| 15.08.-15.09.14 | *Cc, Dol, Pal (5 wt. %) | **PSA 2:** Aaiun-Tarfaya Basin | C/K = 0.0–1.0 *Pal: 1-30 wt% | Limestone deposits [6] Outcrops near Laâyoune with dolomites [6] |
| **CBi** | | | | |
| 26.02.-18.03.14 | *Chl, Kao (C/K = 1), Pal (11 wt. %), Flu, Anh, Sme, Ga | **PSA 2:** Aaiun-Tarfaya Basin near Boucraa | C/K = 0.0–1.0 *Pal: 1-30 wt% | Phosphate deposits [7] |
| 16.12.14-04.01.15 | *Kao (C/K = 0), Pal (1 wt. %), Zeo, Se, Sme | dike swarms and sills of northern Taoudeni Basin | - | Basalts with glass [9] |
| | | **PSA 2:** Aaiun-Tarfaya Basin | C/K = 0.0–1.0 *Pal: 1-30 wt% | Palygorskite-sepiolite mafic clays [7] |
| | | Southern Senegal-Mauritania Basin | - | Lateritic soil [8] Horizontal layers of palygorskite and sepiolite [8] |
| 25.06.-16.07.13 | *Chl, Pal (2 wt. %), Rut | **PSA 2:** Mauritanides | C/K = 0.0–1.0 *Pal: 1-30 wt% | Strongly metamorphosed rocks [10] Greenschist facies [11] |
| 01.08.-21.08.14 | *Fe-Amf, Zeo | dike swarms and sills of northern Taoudeni Basin | - | Basalts with glass [9] |
| | | **PSA 4:** Fezzan uplift | C/K = 0.0–2.6 *Pal: 0 wt% | Zeolite in basaltic rocks [12,13] |
| | | **PSA 3:** Pharusian belt | C/K = 0.2–0.9 *Pal:1-5 wt% | Blueschists [14] Glaucophane bearing eclogites [15] |

* Amf = amphibole, Pal = palygorskite, Chl = chlorite, Cc = calcite, Dol = dolomite, Gib = gibbsite, Zeo = zeolite, Kao = kaolinite, Sme = smectite, Se = sepiolite, Rut = rutile, Serp = serpentine, Ga = garnet, Anh = anhydrite, Flu = fluellite

[1] Schofield et al. (2006) and references therein [2] Schlüter (2008) [3] Wissmann (1982) [4] Bertrand-Sarfati et al. (1991) [5] Biscaye (1964) [6] Bosse and Gwosdz (1996) [7] Moreno et al. (2006) [8] García-Romero et al. (2007) [9] Verati et al. (2005) [10] Villeneuve (2005), [11] Dallmeyer and Lécorché (2012) [12] Abdel-Karim et al. (2013) [13] Cvetković et al. (2010) [14] Caby (2014), [15] Caby et al., 2008) [16] Scheuvens et al. (2013)

We removed the last sentence in the summary and conclusions:

'Finally, we have shown how the mineralogical composition of the samples can be used for provenancing of dust particles found in both on-land and marine dust archives.'

**Specific comments:**

**Comment of Referee #1:**

Line 67-69: Is this statement taken from one of the papers cited in the text close by? Which one? And in general: How far offshore can regional / local dust emissions be found? (Or, in other words, how large is the "footprint" of a dust source when it does not emit the dust to large heights?)

**Reply to comment:**

*This is a good point. This sentence was formulated unclearly. It was related to the findings of the cited papers in the sentences before so we modified the sentence. Unfortunately, the exact distance to which regional/local dust emissions can be found offshore is not known. We hope to resolve this question in the future by comparing dust that is collected with three boys offshore.*

**Changes made in manuscript:**

We modified the following sentence:

Page 2, lines 66-68:

'The results of the above mentioned studies imply that dust collected on land is predominantly of local provenance, while the sources of dust sampled offshore NW Africa are of regional and long-distance provenance.'

**Comment of Referee #1:**

Line 82: Do you mean your Fig. 1 or the Fig. 1 from Scheuvens et al.? Clarify.

**Reply to comment:**

*This was related to the Fig. 1 of this manuscript.*

**Changes made in manuscript:**

We modified the sentence to:

Page 3, lines 82 - 90:

'In Fig. 1 the location of the study sites and the North African dust sources are displayed. The buoy 'Carmen' (~21°15' N, ~20°56' W) and the sediment trap mooring sites CB (~21°16' N, ~20°48' W) and CBi (~20°45' N,

~18°42' W) are located ~ 200 and ~ 80 nautical miles offshore Cape Blanc in the north-eastern (NE) equatorial Atlantic Ocean. The continental dust collector Iwik (~19°53' N, ~16° 18' W) and the meteorological station Arkeiss (~20° 7' N, ~16° 15' W) are located in a major potential dust source area (**PSA 2**) in the Parc National de Banc d'Arguin (PNBA) near Iwik and near Arkeiss in Mauritania. A further meteorological station is positioned in the **PSA 2** in Nouadhibou (~20° 55' N, ~17° 1' W) in the Western Sahara.

The major PSA of northern African dust are summarized in a review by Scheuvens et al. (2013).'

**Comment of Referee #1:**

Line 383: Arkeiss and Iwik are not oceanic, are they? Rephrase.

**Reply to comment:**

*Yes, you are absolutely right. We removed the word 'oceanic'.*

**Changes made in manuscript:**

We modified the sentence to:

Page 15, lines 399-401:

'Moreover, the TRMM satellite product indicated larger rainfall frequencies during the summer season compared to the winter season regarding the stations Carmen, CBi, Iwik and Arkeiss.'

**Comment of Referee #1:**

Line 432 - 436: This discussion about dust fluxes is confusing, and I suggest to completely remove it, also because (see next comment):

**Reply to comment:**

*Good point. We decided to remove the detailed comparison in the dust fluxes between the continental and oceanic sites as the sampling height and the sampling method complicates the comparison. We only mention the progressive downwind decrease in the dust fluxes from site Iwik to CBi and CB.*

**Changes made in manuscript:**

We removed the following sentences:

'The dust fluxes were about 1000 times smaller at the oceanic sites compared to the continental site. A stronger decrease in the fluxes was observed from site Iwik to CB during summer compared to winter. The variation in the seasonal average dust fluxes were well comparable between the continental and oceanic site CBi. A seasonal trend in the dust fluxes could not be observed for the ocean sites CBi and CB. However, a seasonal trend was observed for site Iwik when taking into account the spring and fall samples.'

**Comment of Referee #1:**

Line 438: Not surprisingly, the sampling heights has influences the results. This prohibits a direct comparison of mass concentrations and such parameters between the different sampling types (on mast and in traps in the water), while comparing shapes of size distributions and dust compositions might be possible. This should be discussed in the text somewhere. It is mentioned at line 607 again, but also not discussed. Based on this, a comparison of fluxes between the stations makes no sense.

**Reply to comment:**

*Yes, good argumentation. The comparison between the fluxes is complicated as described in the answer to the previous comment. Therefore, we excluded the discussion on the dust fluxes. No trend in the modal particle size was observed with sampling height. Therefore, the particle size distributions were most likely not greatly influenced by the sampling height and a comparison between the size distributions of the sampling sites should be possible.*

**Changes made in manuscript:**

We modified the following sentence:

Page 18, lines 452-453:

'The dust fluxes generally decreased with collection height in the mast between 90 and 290 cm (not shown).'

We removed the following sentences:

'The stronger decrease in the dust fluxes from site Iwik to CB during summer compared to winter (Table 4) may be explained in the following. During summer, dust was additionally transported with the trades to the site Iwik, (Fig. 9-12) leading to anomalously higher dust deposition at site Iwik compared to the oceanic sites. Further, the washout of dust during offshore transport may have depleted the atmospheric dust cloud resulting in strongly decreased dust deposition fluxes at site CB compared to site CBi during summer.'

We inserted and modified the following sentences in the discussion:

Page 31, lines 655-661:

'The average horizontal fluxes at site Iwik were ~ 1000 times larger with ~ 100000 mgm$^{-2}$d$^{-1}$ (Table 5) due to the different sampling technique. The MWAC samplers do not measure deposition fluxes but foremost dust concentrations. Only 1% or less drops out of a moving dust cloud within five minutes, hence, the horizontal dust flux is at least ~100 times higher than the dust deposition flux (Goossens, 2008). The fact that the dust fluxes decreased with height (not shown) further complicated a comparison between the sites due to the different sampling heights of the dust collectors (2.90 m at Iwik, versus traps in the water). Therefore, the fluxes between the site Iwik and the offshore sediment trap moorings cannot be compared.

**Comment of Referee #1:**

Line 450: The bi-modal distribution of Iwik 13-14 is shown in Fig. 6c, right? Add this information to the text.

**Reply to comment:**

*One of this bimodal distribution is shown in Fig. 6c. The two other bimodal distributions were measured for a spring and a summer sample and are therefore not included in Fig. 6.*

**Changes made in manuscript:**

We modified the following sentences:

Page 19, lines 464-466:

'The three bimodal distributions of the Iwik 13-14 time series were characterized by an additional fine mode peaking at ~16 μm besides the more pronounced and variable coarse mode peaking at ~42 to 55 μm. The three Iwik dust samples characterized by a fine grain-size peak were collected during spring, summer (Fig. 6c) and fall.'

**Comment of Referee #1:**

Fig. 6: There is a change in sequence between the two captions, which I found disturbing.
I'd prefer to always list - CBi - CB - Iwik. Also: maybe put the black data on top of red one, in the plot, otherwise they are hard to see, or make a mix, so that at least some of the black ones are visible a bit better.

**Reply to comment:**

*The comment helped to improve the figure. We put the black data on top of the red one in the plot to improve the visibility. Also, you are right, the order of sites in the legend was not consistent. However, we suggest to present the site CB first – as it is the most distant one – and the site Iwik last – as this is the most proximal sampling site. Or in other words: we displayed the sites in the legend now from West to East.*

**Changes made in manuscript:**

We modified figure 6:

Page 19, line 480:

[Figure]

**Comment of Referee #1:**

Line 452: Explain what you mean by "the sorting".

**Reply to comment:**

*The sorting is expressed by the standard deviation of the grain-size distribution: the larger the standard deviation the weaker the sorting. We added this explanation to the text.*

**Changes made in manuscript:**

We inserted the following sentence:

Page 19, lines 468-469:

'The standard deviation of a grain-size distributions is a measure of the sorting of the dust sample: the larger the standard deviation the weaker the sorting.'

**Comment of Referee #1:**

Figure 7 a) and b): Why was a different wind speed chosen in a) (> 5 m/s) and b) (> 6.5 m/s)? Also: there is a frame around Fig. 7 (on 3 of the 4 sides) that has to be removed for the final version.

**Reply to comment:**

*The explanation on why we chose these values is missing in the text. Only the threshold wind speed of the wind events for which the correlation was best was shown in the graph. This optimum threshold wind speed was different between the correlation to the dust fluxes and the correlation to the modal particle size. We added this information to the revised manuscript.*

**Changes made in manuscript:**

We inserted the following sentences:

Page 20, lines 488-489:

'The correlation was only evident when using a threshold for wind events of 3.5 to 5.5 ms$^{-1}$ and was best for a threshold of 5 ms$^{-1}$.'

Page 20, lines 494-495:

'The correlation was only evident when using a threshold for wind events of 6.5 to 7 ms$^{-1}$ and was best for a threshold of 6.5 ms$^{-1}$.'

We modified fig. 7:

Page 20, line 502:

[Figure]

**Comment of Referee #1:**

Line 527: Why are there 2 back trajectories, only? Were they only made for the time of the dust event? Then why 2 and not only 1? This has to be clearly explained (this also is valid for Fig. 10, 11, and 12, where different numbers or trajectories are shown.)

**Reply to comment:**

*A backward trajectory was drawn only for the day with a dust storm event. So in figure 9 of the old version of the manuscript two days with a dust storm event were identified during the sampling interval of each site resulting in two back trajectories per height at each site. We inserted this information to the text.*

**Changes made in manuscript:**

We inserted the following sentences:

Page 22, lines 547-548:

'A back trajectory was drawn for the day when a dust storm event occurred as depicted on the satellite images.'

Page 23, lines 553-556:

'During the sampling interval of each site at least two days with dust storms occurred (Fig. 9c,d). Therefore, two back trajectories were drawn for each height for the site CBi and CB respectively. The high-level back trajectories ending at site CBi pass either through the major **PSA 2** or point offshore.'

Page 24-25, lines 568-571:

'During the sampling interval of the site Iwik at least three dust storms occurred and at the site CBi at least two dust storms occurred (Fig. 10f-h). Each dust storm lasted for several days for which we could model as many as 15 back trajectories for the site Iwik and 8 for the site CBi for each height. The large number of back trajectories complicated the determination of the likely source areas.'

Page 26-27, lines 590-592:

'Only one dust storm event was observed during the sampling interval at both sites which lasted for one day (Fig. 11c) resulting in only one back trajectory per site and per height.'

Page 28, lines 603-607:

'At least five separate dust events could be identified (Fig. 12f-j) out of which three occurred during the sampling interval of the site Iwik and two during the sampling interval of the site CBi. One of these dust storms occurring during the sampling interval of site CBi lasted for two days (07-08.08.2014), while all other dust storms lasted for only one day. As a result, three back trajectories could be drawn for each site and each height.'

**Comment of Referee #1:**

Line 538: Just out of curiosity: does sea spray (i.e., the Cl in it) engrave the analysis of chloride?

**Reply to comment:**

*No, because the CBi samples were pre-treated with distilled water before analysis resulting in the washout of Cl-ions.*

**Comment of Referee #1:**

Figure 9: The line for the 4500m back-trajectory for CBi is in the caption, but not in the plot. But it could be interesting to see it in here, as the dust transported from further away at higher altitudes might also contribute. This comment is valid also for Fig. 10.

**Reply to comment:**

*This was a very good comment which helped to determine the likely source areas of the sampled dust. We added the back trajectories at 3000 and 4500 m to Fig. 9 and 10. With the new Fig. 10 we could find a new source area in Senegal which may have supplied the characteristic minerals kaolinite and sepiolite present in the sample.*

**Changes made in manuscript:**

Page 25, lines 582-584:

'The presence of the mineral kaolinite in this marine winter sample may be explained by a kaolinite-rich source area lying in the southern Senegal-Mauritania Basin (Fig. 10e).'

**Comment of Referee #1:**

Line 551: Replace "due to the" by ""as there are".

**Reply to comment:**

*We revised the sentence.*

**Changes made in manuscript:**

We modified the sentence to:

Page 25, lines 576-578:

'Chlorite may have been supplied to Iwik from a source area nearby the Senegal-Mauritania Basin (Fig. 10a) or as far as the eastern Taoudeni Basin (Fig. 10b) as there are anomalously high chlorite content of the soils in these areas.'

**Comment of Referee #1:**

Line 596 ff: Was this for PM10 or total particulate mass? What exactly do you mean by "annual average dust concentrations"? The average per dust storm, or really the all-time average of dust, or something else? Please explain. Also: It is not clear to me how the monitoring of less dust events in Morocco due to the shorter sampling time can affect an average value, unless this average is "per year".

**Reply to comment:**

*The annual average dust concentrations were calculated for total particulate mass using all dust samples. We modified the sentence to explain this. We also noticed that the sentence on the monitoring of dust events was not adequate also because the average value was calculated by excluding haze-periods and dust-storms at the sampling site. Therefore, the statement was removed and we modified the paragraph.*

**Changes made in manuscript:**

We removed the sentence:

'However, in Morocco dust was collected at a larger height of 4 m and the sampling time is much shorter leading to the monitoring of less dust events. '

We modified the sentences to:

Page 31, lines 640-644:

'An annual average dust concentration (total suspended particles) of ~214 $\mu gm^{-3}$ and 275 $\mu gm^{-3}$ was estimated for all dust samples of the year 2013 and 2014 respectively regarding the site Iwik (Table 4). These estimates were larger than what has been measured for background dust concentrations (total suspended particles) in Morocco which were in the order of 100 $\mu gm^{-3}$ during spring 2006 (Kandler et al., 2009). However, in Morocco dust was collected at a larger height of 4 m and haze-periods and dust-storms were excluded from the average value.'

**Comment of Referee #1:**

Line 606-607: If the difference is due to different sampling techniques, does it make sense to do such a comparison as you present it here?

**Reply to comment:**

*You are right. A comparison of the dust fluxes between the onshore and offshore sites is not feasible because of the different sampling techniques. Therefore, we excluded the comparison of the fluxes between site Iwik and the oceanic sites CB and CBi in the discussion.*

**Changes made in manuscript:**

We modified and shifted the following paragraphs to:

Page 31, lines 651 – 661:

The observed general decrease in the dust flux from the sites CBi and CB can be explained via the increase in the distance to the source area. Decreased dust deposition fluxes offshore NW Africa with increasing distance from the African coast were also observed by Bory and Newton (2000) analysing the lithogenic fluxes in marine sediment traps.

The average horizontal fluxes at site Iwik were ~ 1000 times larger with ~ 100000 mgm$^{-2}$d$^{-1}$ (Table 5) due to the different sampling technique. The MWAC samplers do not measure deposition fluxes but foremost dust concentrations. Only 1% or less drops out of a moving dust cloud within five minutes, hence, the horizontal dust flux is at least ~100 times higher than the dust deposition flux (Goossens, 2008). The fact that the dust fluxes decreased with height (not shown) further complicated a comparison between the sites due to the different sampling heights of the dust collectors (2.90 m at Iwik, versus traps in the water). Therefore, the fluxes between the site Iwik and the offshore sediment trap moorings cannot be compared.

**Comment of Referee #1:**

Line 608: In which time span does 1% drop out of a moving dust cloud? Per day, per year, ever (certainly not), : : :?

**Reply to comment:**

*In a time span of five minutes. We changed the sentence accordingly.*

**Changes made in manuscript:**

We modified the sentence to:

Page 31, lines 657-658:

'Only 1% or less drops out of a moving dust cloud within five minutes, hence, the horizontal dust flux is at least ~100 times higher than the dust deposition flux (Goossens, 2008).'

**Comment of Referee #1:**

Line 613-614: Replace "in the following" by "as follows".

**Reply to comment:**

*We excluded the sentence according to the comments on the fluxes above.*

**Changes made in manuscript:**

We removed the paragraph:

'The stronger decrease in the dust fluxes from site Iwik to CB during summer compared to winter (Table 4) may be explained in the following. During summer, dust was additionally transported with the trades to the site Iwik, (Fig. 9-12) leading to anomalously higher dust deposition at site Iwik compared to the oceanic sites. Further, the washout of dust during offshore transport may have depleted the atmospheric dust cloud resulting in strongly decreased dust deposition fluxes at site CB compared to site CBi during summer.'

**Comment of Referee #1:**

Line 638: You could stress stronger that a bi-modal size distribution (as observed e.g., at Iwik in Fig. 6c) be indicative for nearby sources. Different size modes typically indicate different sources!

**Reply to comment:**

*Yes, of course bimodal distributions are indicative for different sources. Since this has not been stated clearly enough, we further stressed this in the manuscript. Bimodal distributions may further be indicative of precipitation in the study area due to the deposition of fine particles from higher altitude. This is discussed in the text in chapter 4.1.2.*

**Changes made in manuscript:**

We inserted the following sentence:

'Bimodal grain-size distributions typically indicate the sampling of different dust sources (Stuut et al. (2009) and references therein).'

**Comment of Referee #1:**

Line 641: Not only the wind speed, but also the heights into which dust was emitted, will influence how far it can be transported.

**Reply to comment:**

*Absolutely right. We modified this in the manuscript.*

**Changes made in manuscript:**

We modified the following sentence:

'Therefore, it may be possible that wind velocities were high enough during the sampling interval to inject dust to higher altitude and transport it from more distant sources (Fig. 12b) to the sampling site resulting in the small peak in the grain-size distributions.'

**Comment of Referee #1:**

Line 648: I have a hard time imagining how precipitation droplets (with typically high fall velocities) enter your sampler in noticeable amounts. Or do you suggest here that precipitation is formed, falls, evaporates during its fall and leaves these dust particles behind at lower altitudes which can then be sampled? If this is so, mention it. This is only partly true as there were clearly remnants of water in the bottles. So, both are true!

**Reply to comment:**

*Yes, thanks, it could be a good explanation that precipitation evaporates during its fall leaving the dust particles behind at lower altitude. So we added this to the text. However, we also observed that there were clearly remnants of water in the bottles. So also wet deposition into the bottles may have occurred!*

**Changes made in manuscript:**

We inserted the sentence:

'The rain droplets may have evaporated during their fall releasing the dust particles at lower altitudes which can then be sampled with the MWAC sampler. However, we also observed some remnants of water in the bottles and therefore wet deposition into the bottles may have also occurred.'

**Comment of Referee #1:**

Line 692: Prior to this line, you explain that quartz seems to be present "everywhere" in North Africa. If this is the case, a high quartz content cannot be used to assume that it is mainly locally derived dust.

**Reply to comment:**

*Good point. We removed the sentence.*

**Changes made in manuscript:**

We removed the sentences:

'Moreover, the continental sampling site is surrounded by sand dunes which are rich in quartz minerals (Schlüter, 2008;Lancaster, 2013). A high quartz content may therefore point to predominantly locally derived dust.'

**Comment of Referee #1:**

Line 736: This argument can only be correct if only this one nearby source does not have the mineral palygorskite in it, while it would be present in ALL other sources. Also: not finding palygorskite might simply be an issue of low sampling statistics – can you exclude this? So overall: Can this text in the manuscript really be stated like this? This is also connected to what you state in line 709, so check this location for consistency, too.

**Reply to comment:**

*We revised the sentences in the manuscript in which this was stated incorrectly. Palygorskite may have been derived from several local sources in which it was not present in the soils instead of a single localized source.*

**Changes made in manuscript:**

We modified the sentence:

'Therefore, we argue that the sampled dust was most likely derived from a localized source of **PSA 2**.'

Page 35, lines 793:

'This may point to several externally mixed sources of **PSA 2** during transport.'

Page 36, lines 807-808:

'Again, the absence of the mineral palygorskite is noteworthy which may point to the sampling of a localized dust source.'

Page 36. lines 820-821:

'Hence, dust may have been supplied from several dust sources of **PSA 2** which were mixed during transport.'

**Comment of Referee #1:**

Line 784 ff: How fast do dust particles sink in the water – can they travel 500km and still be collected at the sampling site CBi? Or, in other words, how large is the catchment area of a trap? You mention 40km * 40 km above in the text yourself for the upper trap? Therefore, it seems that the here described time delay cannot be used as an explanation.

**Reply to comment:**

*We argued that certain clay minerals may have been transported with the undercurrent to the sampling site. When not incorporated into marine aggregates clay minerals can easily be transported over long distances within nepheloid layers (ca. 500 km) because of their small size.*

**Comment of Referee #1:**

Line 833-835: For continental sampling, rather than using the word "trade wind", the word "nearby sources" would be more appropriate. Also, the discrimination between dust particles transported in the SAL or by the trade winds is awkward. The trade winds transport the SAL, while the SAL is the source of particles that deposit from air masses that are moved by the trade winds. In this sense, "trade wind" and "SAL" both contribute together and should not be separated. The formulation needs rewording. Also check the whole text to remove respective inconsistencies.

**Reply to comment:**

*We agree that the sentence should be reformulated.*

**Changes made in manuscript:**

We modified the sentence:

Page 39, lines 909-911:

- 'dust deposited on the continent was predominantly transported from near-by local sources (Mauritania, Western Sahara and Mali), while dust deposited in the marine traps was transported from proximal (Mauritania, Western Sahara and Mali) and distal sources (Algeria and Libya)'

**Comment of Referee #1:**

Line 836-837: This result was only presented in bypassing (line 731 – or am I missing something), and I was quite surprised to see this as one of the main findings. On the other hand, changing grain sizes with distance to sources are not mentioned at all. I suggest to really reconsider which of your results are worth mentioning here, as for some readers, abstract and summary might be all they will ever look at.

**Reply to comment:**

*Yes, you are right, this was not one of the major findings so we removed it from the summary and conclusions. The downwind decrease in particle size is mentioned in the discussion on page 32, lines 676-682 and pages 33-34 lines 752-756.*

**Changes made in manuscript:**

We removed the sentence:

- 'the percentage of mica relative to the quartz content increased in the deposited dust with increasing transport distance, most likely due to the platy shape of these minerals, which reduces settling'

**Comment of Referee #1:**

Line 842 ff: The wind at the sampling location might differ from the wind in the source region, if the latter is far away, and the sizes of dust particles present in the source region will influence the grain sizes that can be deposited as well. There might still be a connection between wind speed and transported grain size, but it should be discussed in a broader sense, including (or at least mentioning) the points I raise in the previous sentence.

**Reply to comment:**

*We included these considerations in the summary and outlook of the revised manuscript.*

**Changes made in manuscript:**

We inserted the sentence:

Page 39, lines 919-922:

[revised manuscript text omitted]

**Anonymous Referee #2**

In the manuscript, "Seasonal provenance changes of present-day Saharan dust collected on- and offshore Mauritania" the authors present data from sediment traps at multiple depths off the coast of Mauritania and surface collection sites close to the coast. They attempt to determine location and seasonality in potential source regions for different case studies based on the mineralogy of the samples and back trajectory analysis.

**Comment of Referee #2:**

The description of the methodology and measurements is comprehensive and well thought out. I cannot speak to the specifics of the measurement methodology but have added minor comments and clarification requests below. Unfortunately, the broader context and the scientific developments are lacking in the paper. The measurements are clearly valuable and should be published. The analysis of collected samples and the potential source regions is thorough in a qualitative sense. However, the useful scientific conclusions are not clear. This is indicated by the abstract that reads more like and introduction, the long, subjective discussion, and the relatively sparse summary and conclusions. For example, the last paragraph of the manuscript states that sediment records from land and ocean are likely to sample different source regions, based on the measurements showing more local sources over land. This could be an interesting point, but without further analysis (firmer understanding of the sources, dependence on the particular measurement site) the conclusion that sources are more likely to be local on land than at an ocean site further downwind seems common sense. I'm also not convinced that the atmospheric and sediment trap data should be presented side by side based on the difference in collection methodology and catchment. I think the authors need to consider how to better frame the important measurements presented in this manuscript, by presenting the data in a way that is easier to compare with other dust deposition and concentration measurements and a more thorough back trajectory analysis that answers the questions laid out in the introduction.

**Comment of Referee #2:**

The choice of questions (lines 76-79) is a little strange. For example, (1) why would one not expect there to be seasonality in the deposition when we know that there is seasonality in winds leading to dust emission and transport? (2) This is an interesting question, but how dependent is this on the specific locations chosen? (3) This is very similar to question (1). (4) This is a good question but is only tackled in a qualitative way in the manuscript. I think laying out the questions to be answered is a good format; however, they currently seem like an afterthought and they should be returned to explicitly in the summary/conclusions.

**Reply to comment:**

*Yes, it is correct that the questions were not formulated and addressed adequately in the old version of the manuscript. Therefore, we modified the research questions. Further, we answered them more precisely in the*

*discussion, summary and conclusions part of the revised manuscript. Especially the newly developed table 6 which was demanded by reviewer 1 should help to answer the revised research question 2-3.*

**Changes made in manuscript:**

We modified the research questions (page and line numbers refer to the revised manuscript):

Page 3, lines 76-79:

[revised manuscript text omitted]

**Comment of Referee #2:**

While the back trajectory analysis is interesting, it is rudimentary. The choices of back trajectory heights appear to be arbitrary, I did not find a reason for the choice of 4500 m and 10 m is understandable, but those trajectories will be highly uncertain. A larger ensemble of back trajectories from different altitudes and start points would better quantify the likelihood of dust sources and also help represent the uncertainties in back trajectories that pass so close to the surface.

**Reply to comment:**

*This comment was raised by reviewer #1 as well. We have choosen the altitudes for back trajectories on the basis of previous studies to which we compared our data. For example, Skonieczny et al. (2013) used 4500m to study the SAL. Following the reviewer's suggestion, we now also included additional back trajectories, which made the observed patterns much clearer. Back-trajectories of the heights 100 m and 3000 m were added to figures 9-12. This improved the detection of the likely dust source areas and the results and interpretations were adjusted accordingly.*

**Changes made in manuscript:**

We modified fig. 9-12:
Page 24, line 563:

[Figure]

**Lithology**

- Carbonate
- Karst
- Non-Carbonate
- Metasedimentary
- Alkaline Intrusive Volcanic
- Silicic
- Metaigneous
- Ultramafic
- Extrusive Volcanic
- Colluvium
- Hydric - Organic
- Aeolian Sediments
- Alluvium - Fan Deposit
- Alluvium - Fluvial
- Alluvium - beach, strand, coastal dune
- Alluvium - Saline
- Alluvium - Gypsum
- Alluvium - Other
- Volcanic - Ash, Tuff, Mudflow
- Water

**Back trajectories**

- Iwik
- CBi

Back trajectories: 10 m, 100 m, 10 m, 100 m, 3000 m, 4500 m

**PSAs:** PSA 1-5

**Clay Mineralogy**

- No data
- Calcite >8.9 %
- Kaolinite >29 %
- Chlorite >4.1 %

**Mineralogy**

- Kaolinite
- Chlorite
- Calcite
- Quartz
- Feldspar
- Mica
- Amphibole
- Palygorskite
- Serpentine
- Rutile
- Zeolite
- Fluellite
- Sepiolite
- Anhydrite
- Smectite
- Garnet
- Gibbsite
- Dolomite
- Other minerals
- Ferryglaucophane (c) 28.02.2014

(d) 01.03.2014

[Figure]

[Figure]

**Lithology**

| | | |
|---|---|---|
| Carbonate | Ultramafic | Alluvium - beach, strand, coastal dune |
| Karst | Extrusive Volcanic | Alluvium - Saline |
| Non-Carbonate | Colluvium | Alluvium - Gypsum |
| Metasedimentary | Hydric - Organic | Alluvium - Other |
| Alkaline Intrusive Volcanic | Aeolian Sediments | Volcanic - Ash, Tuff, Mudflow |
| Silicic | Alluvium - Fan Deposit | Water |
| Metaigneous | Alluvium - Fluvial | |

**Back trajectories**

● Iwik   ● CBi

— 10 m   — 100 m   — 10 m   — 100 m   - - - 3000 m   - ■ - 4500 m

**PSAs:** ☐ PSA 1-5

**Clay Mineralogy**

▨ No data   ▨ Calcite >8.9 %   ▨ Kaolinite >29 %   ▤ Chlorite >4.1 %

**Mineralogy**

| | | | | |
|---|---|---|---|---|
| Kaolinite | Feldspar | Serpentine | Sepiolite | Gibbsite |
| Chlorite | Mica | Rutile | Anhydrite | Dolomite |
| Calcite | Amphibole | Zeolite | Smectite | Other minerals |
| Quartz | Palygorskite | Fluellite | Garnet | ★ Ferryglaucophane |

(c) 02.07.2013

Page 29, line 616:

[Figure]

We modified the following sentence to:

Page 14, lines 373-377:

'Four-day back trajectories at altitudes of 10 (following Stuut et al. (2005)), 100, 3000, 4500 (following Skonieczny et al. (2013)) and 5500m were calculated ending at the dust collector site Iwik (19°52' N, 16°17' W) and at the proximal marine trap site CBi (20°46',18°44' W) using the Hybrid Single Particle Langrangian

Integrated Trajectory (HYSPLIT) model (Stein et al., 2015) and the reanalysis dataset (2.5° spatial resolution) on the NOAA website at http://ready.arl.noaa.gov. '

We modified the following sentences:

Page 22, lines 546-547:

'Four heights, 10, 100, 3000 and 4500 m were chosen to cover both low- (trades) and high-level (SAL) dust transport.'

Page 22, lines 551-552:

'The back-trajectories at 5500 m can be found in the supplements. '

Page 23, lines 561-562:

'Thus, the source area of the samples was most likely the chlorite and kaolinite rich sediments located near the Bou Craa phosphate mine in the Western Sahara (Fig. 9b).'

Page 25, lines 581-584:

'A further source area might be the southern shoreline of the Western Sahara in which chlorite depleted sediments are situated (Fig. 10d). The presence of the mineral kaolinite in this marine winter sample may be explained by a kaolinite-rich source area lying in the southern Senegal-Mauritania Basin (Fig. 10e).'

**Comment of Referee #2:**

Further, including surface wind reanalyses (and other meteorology) in a comprehensive analysis of the mineralogy measurements and back trajectories, a more rigorous statistical analysis of likely source regions could be undertaken. A statistical approach would also provide a framework to analyze future measurements, rather than the case-by-case methodology shown. This would also help reduce the speculatory nature of the discussion of sources in Section 4.2.

**Reply to comment:**

*Very good point. We followed your advice and executed an analysis of surface wind reanalysis data (20[th] century reanalysis V2c dataset). This analysis proved anomalously high surface wind velocities for the respective dust storm events and chosen dust source areas. Therefore, the analysis supported the derivation of the source regions based on mineralogy.*

**Changes made in manuscript:**

We inserted the following sentences:

Page 13, lines 362-364:

'Maps of six hourly mean surface wind vectors and speed (20[th] century reanalysis V2c dataset) were provided by the NOAA/OAR/ESRL PSD, (Boulder, Colorado, USA) and downloaded from their website (http://www.esrl.noaa.gov/psd/). '

We inserted the following paragraph and figure:

Page 29-30, lines 621-632:

'In Fig. 13a-d the mean wind vectors and speed are presented for chosen dust storm events. The individual dust source areas that were identified using the back trajectory of the day with the dust storm as shown in Fig. 9-12 are further displayed in Fig. 13a-d. As can be clearly seen in the subfigures, the mean wind velocities were anomalously large in the chosen dust source areas which enabled dust emission. During winter, six hourly mean wind velocities were larger than 7 ms$^{-1}$ in the chosen dust source areas (Fig. 13a-b). During summer 2013, six hourly mean wind velocities were larger than 6 ms$^{-1}$ in the chosen dust source area (Fig. 13c). During summer 2014 extremely high mean wind velocities were encountered near the study sites and in the dust source area enabling dust emission and transport from a more distant source to the site Iwik (Fig 13d).

[Figure]

**Figure 13: Six hourly composite mean wind vectors and speed at 1000 mb for selected days including a dust storm event during winter (a) – (b) and summer (c) – (d). The dust source area that was identified for the individual dust storm event using the back trajectory of the day with the dust storm is further displayed.'**

We inserted the sentence:

Page 32, lines 689-691:

'This interpretation is further supported by the reanalysis wind vector maps showing anomalously high wind velocities between the site Iwik and the proposed distant source area enabling dust emission and transport of dust particles from the more distant source to the site Iwik (Fig. 13d).'

**Comment of Referee #2:**

Are the distributions really unimodal in Figure 6, as stated on line 620? I can see the finer mode and sometimes a third mode in there.

**Reply to comment:**

*You are right. Some distributions have a finer mode. However, most of the distributions are unimodal. This may have been formulated unclearly in the old version of the manuscript. Therefore, we modified the sentence.*

**Changes made in manuscript:**

We modified the sentence to:

Page 31, lines 664-665:

'The measured grain-size distributions for dust trapped at 2.90 m on land at site Iwik and for dust settling in the ocean were predominantly unimodal (Fig. 6).'

**Comment of Referee #2:**

Coming from an atmospheric modeling perspective, the mode between 1-10um is of great interest and it is a shame that this is not discussed more. From an atmospheric perspective, the value of this work could be increased by presenting more information on the finer dust particles. Models are always in need of aerosol size distribution measurements for evaluation of dust emission, transport and deposition.

**Reply to comment:**

*We also agree that we should describe the modes of the offshore dust samples more in detail as we have described the fine mode of the Iwik samples. Therefore, we added a more detailed description in the result section. The reason for the fine mode which we observe in the CBi samples may be mainly due to different transport distances and source areas as has been already stated in the discussion part of the old manuscript on page 32, lines 699 -701.*

**Changes made in manuscript:**

We inserted the sentences:

Page 19, lines 467-468:

'The eight bimodal grain size distributions of the CBi 11-12 time series were characterized by a variable coarse mode at ~ 25 to 35 μm and a variable fine mode at ~ 6 to 16 μm.'

**Comment of Referee #2:**

It would also be useful to have the size distributions presented as dM/dlnD or dV/dlnD to allow for comparison with model simulations and other measurements.

**Reply to comment:**

*Good point as many modellers read and publish in the journal ACP. Therefore, we followed your suggestion and revised the figure 6.*

**Changes made in manuscript:**

We modified figure 6:

Page 19, lines 480:

[Figure]

**Comment of Referee #2:**

To summarize, I think:

- The measurements and presentation of the results are of great value

- Determining dust sources could benefit greatly from an analysis of surface winds from meteorological reanalyses to accompany the back trajectories.

- The summary and conclusions seems like an afterthought in the current presentation.
Consider expanding this to crystallize the findings of the research better.

- The questions to be answered that are set out in the introduction are vague. It is useful to set out the motivational questions for the manuscript in this way, but I think more time should be put into the questions to be addressed and then ensuring that you return to these points in the discussion/summary/conclusions.

**Reply to comment:**

*Since this comment summarizes the above mentioned comments, we have already answered them above.*

**Minor Comments**

**Comment of Referee #2:**

line 14-15, 34 "Environmental parameters" is non-descriptive. Please revise. Line 34 could be deleted.

**Reply to comment:**

*We revised the sentences to clarify what we mean by environmental parameters. Further, we also agree that line 34 could be deleted.*

**Changes made in manuscript:**

We modified the sentences:

Page 1, lines 13-16:

'Saharan dust has a crucial influence on the earth climate system and its emission, transport, and deposition are intimately related to e.g. wind speed, precipitation, temperature and vegetation cover. The alteration in the physical and chemical properties of Saharan dust due to environmental changes is often used to reconstruct the climate of the past.'

**Comment of Referee #2:**

line 95 "In the following," - add comma

**Reply to comment:**

*We revised this sentence.*

**Changes made in manuscript:**

Page 4, lines 107:

'In the following, the lithology of the geological provinces that underlay the major PSA's is outlined (Fig. 1). '

**Comment of Referee #2:**

line 131 - Haboobs are normally defined as the dust storm from evaporatively driven cold pool outflow from convective events, not low-level jets

**Reply to comment:**

*This was formulated unclearly. Of course, haboobs are not low-level jets. The sentence included an enumeration of dust emission mechanism. To prevent confusion, we excluded the 'so called' from the sentence.*

**Changes made in manuscript:**

We revised the sentence to:

Page 5, lines 144-145:

'Dust emission is driven by low level jets, 'haboobs', African easterly waves (AEWs) and high surface winds associated with the Saharan heat low (Knippertz and Todd, 2012).'

**Comment of Referee #2:**

line 161-163 - this paragraph seems disconnected from the rest of the section

**Reply to comment:**

*This comment is similar to the comment of reviewer # 1 who asked for a paragraph and figure in which all study sites are introduced together. Therefore, we shifted this paragraph to the introductory sentences of the other study sites and modified it accordingly.*

**Changes made in manuscript:**

We shifted and modified the sentences to page 3, lines 82-88:

'In Fig. 1 the location of the study sites and the North African dust sources are displayed. The dust-collecting buoy 'Carmen' (~21°15' N, ~20°56' W) and the sediment trap mooring site CB (~21°16' N, ~20°48' W) are virtually at the same position ~200 nautical miles offshore Cape Blanc. Sediment-trap station CBi (~20°45' N, ~18°42' W) is located ~ 80 nautical miles offshore Cape Blanc. The continental dust collector Iwik (~19°53' N, ~16° 18' W) and the meteorological station Arkeiss (~20° 7' N, ~16° 15' W) are located in a major potential dust source area (**PSA 2**) in the Parc National de Banc d'Arguin (PNBA) near Iwik and near Arkeiss in Mauritania. A further meteorological station is positioned in the **PSA 2** in Nouadhibou (~20° 55' N, ~17° 1' W) in the Western Sahara. '

**Comment of Referee #2:**

line 222 and 235 repeat

**Reply to comment:**

*We merged the two sentences.*

**Changes made in manuscript:**

We removed the sentence:

'The masts were aligned to the ambient wind direction via a wind vane (Fig. 2).'

We modified the sentence:

Page 10, lines 256-257:

'The masts of the buoy Carmen and of the Iwik dust sampler were aligned to the ambient wind direction via a wind vane (Fig. 2).'

**Comment of Referee #2:**

line 247 - why is 2xCorg removed from the total mass? Is this a general scaling from organic carbon to organic mass?

**Reply to comment:**

*Yes. About 50-60 % of marine organic matter is constituted by organic carbon. Therefore, we estimated the organic matter by multiplying organic carbon by a factor of two.*

**Comment of Referee #2:**

line 350 - I think the long url links should go in the data availability section rather than in the text. Write out the usage but simply reference the data section rather than talking about downloading files in the manuscript.

**Reply to comment:**

*Good point as this will improve the readability of the manuscript. We modified the section '2.8 Mapping with ArcMap' by removing the long url links.*

**Changes made in manuscript:**

The following sentences were modified to:

Page 14, lines 378-382:

'An ArcGIS layer file of the African surface lithology (new_af_lithology_w_glbcvr_waterbdy_90m_dd84_final.lyr) was downloaded from the U.S. Geological survey (USGS) website (http://rmgsc.cr.usgs.gov).

An ArcGIS shape file of the African soils (DSMW.shp) was downloaded from the website of the food and agriculture organization of the United Nations (FAO) (http://www.fao.org).'

**Comment of Referee #2:**

line 357-358 - this repeats the previous sentence, condense.

**Reply to comment:**

*We condensed the two sentences into one sentence.*

**Changes made in manuscript:**

The sentence was modified to:

Page 14, lines 382-384:

'The mean percentage of calcite (8.9 %), chlorite (4.1 %) and kaolinite (29%) in the clay fraction of Saharan soils in general and for each soil type is given by Journet et al. (2014).'

**Comment of Referee #2:**

line 377 - "ground station" could be misinterpreted as on land, consider "surface station"

**Reply to comment:**

*We changed the word 'ground station' into 'surface station' all through the revised manuscript.*

**Changes made in manuscript:**

Page 4, lines 98-100:

**Figure 1: Map of the study sites under investigation: the scientific buoy Carmen as well as the sediment trap moorings CB and CBi offshore Cape Blanc, the MWAC dust collector onshore near Iwik and the surface stations near Nouadhibou and Arkeiss**

Page 15, lines 395-396:

'Regarding the surface stations Carmen and Arkeiss, a threshold of >0.2 mmd$^{-1}$ was used in order to exclude events which may be related to anomalously high moisture instead of rainfall.'

Page 15, lines 405-407:

'However, the spatial and seasonal trends observed by the TRMM data were not supported by the sensor on buoy Carmen and by the surface station in Arkeiss.'

Page 15, lines 408-410:

'Further, disagreements between the surface stations and the TRMM stations maybe caused by the local signal recorded by the respective rain sensor.'

Page 16, line 416:

'The wind direction and speed for the surface stations Carmen, Nouadhibou and Iwik are displayed in Fig. 4c.'

**Comment of Referee #2:**

line 461 - define "well sorted"

**Reply to comment:**

*The same comment raised by reviewer # 1. 'Well sorted' refers to the standard deviation of the particle size distribution: the smaller the standard deviation the better the sorting. We added this information to the text.*

**Changes made in manuscript:**

We inserted the sentence:

Page 19, lines 468-469:

'The standard deviation of a grain-size distributions is a measure of the sorting of the dust sample: the larger the standard deviation the weaker the sorting.'

**Comment of Referee #2:**

Figure 6 - These are not really unimodal, as referred in the text (line 620)

**Reply to comment:**

*Most of them are unimodal, while some are bimodal. That was formulated unclearly. So we modified the sentence on line 620 of the old version of the manuscript.*

**Changes made in manuscript:**

We modified the following sentence:

Page 31, lines 664 – 665:

'The measured grain-size distributions for dust trapped at 2.90 m on land at site Iwik and for dust settling in the ocean were predominantly unimodal (Fig. 6).'

**Comment of Referee #2:**

line 602 - which year?

**Reply to comment:**

*That was for the year 2013.*

**Changes made in manuscript:**

The sentence was modified to:

Page 31, lines 647:

'At the distal oceanic site CB, the annual average dust deposition flux was ~ 45 mgm$^{-2}$d$^{-1}$ (2013) (Table 4).'

**Comment of Referee #2:**

line 608 - is horizontal flux a useful metric to compare with ocean deposition?

**Reply to comment:**

*We also realized that it is not possible to compare the dust fluxes measured at site Iwik with the dust fluxes of the sites CBi and CB due to the different sampling techniques and sampling heights. Therefore, we modified the paragraph on the dust fluxes.*

**Changes made in manuscript:**

We modified the following paragraph:

Page 31, lines 655 – 661:

'The average horizontal fluxes at site Iwik were ~ 1000 times larger with ~ 100000 mgm$^{-2}$d$^{-1}$ (Table 5) due to the different sampling technique. The MWAC samplers do not measure deposition fluxes but foremost dust concentrations. Only 1% or less drops out of a moving dust cloud within five minutes, hence, the horizontal dust flux is at least ~100 times higher than the dust deposition flux (Goossens, 2008). The fact that the dust fluxes decreased with height (not shown) further complicated a comparison between the sites due to the different sampling heights of the dust collectors (2.90 m at Iwik, versus traps in the water). Therefore, the fluxes between the site Iwik and the offshore sediment trap moorings cannot be compared.'

**Comment of Referee #2:**

A minor issue, but there are formatting errors with brackets on the references throughout that need fixing.

**Reply to comment:**

*We modified the references which were characterized by formatting errors.*

**Changes made in manuscript:**

Page 2, lines 42-44:

'For instance, the particle size of mineral dust in ocean sediment records varies according to the paleo-frequency of dust-storm and rainfall events (e.g. Friese et al. (2016)).'

Page 2, lines 50-56:

[revised manuscript text omitted]